# CHELSA-TraCE21k – high-resolution (1 km) downscaled transient temperature and precipitation data since the Last Glacial Maximum

**Dirk Nikolaus Karger**[1], **Michael P. Nobis**[1], **Signe Normand**[2], **Catherine H. Graham**[1], **and Niklaus E. Zimmermann**[1]

[1]Swiss Federal Research Institute WSL, Zürcherstrasse 111, 8903 Birmensdorf, Switzerland
[2]Department of Biology, Aarhus University, Ny Munkegade 116, 8000 Aarhus, Denmark

**Correspondence:** Dirk Nikolaus Karger (dirk.karger@wsl.ch)

**Abstract.** High-resolution, downscaled climate model data are used in a wide variety of applications across environmental sciences. Here we introduce a new, high-resolution dataset, CHELSA-TraCE21k. It is obtained by downscaling TraCE-21k data, using the "Climatologies at high resolution for the earth's land surface areas" (CHELSA) V1.2 algorithm with the objective to create global monthly climatologies for temperature and precipitation at 30 arcsec spatial resolution in 100-year time steps for the last 21 000 years. Paleo-orography at high spatial resolution and for each time step is created by combining high-resolution information on glacial cover from current and Last Glacial Maximum (LGM) glacier databases and interpolations using data from a global model of glacial isostasy (ICE-6G_C) and a coupling to mean annual temperatures from TraCE21k (Transient Climate Evolution of the last 21 000 years) based on the Community Climate System Model version 3 (CCSM3). Based on the reconstructed paleo-orography, mean annual temperature and precipitation were downscaled using the CHELSA V1.2 algorithm. The data were validated by comparisons with the glacial extent of the Laurentide ice sheet based on expert delineations, proxy data from Greenland ice cores, historical climate data from meteorological stations, and a dynamic simulation of species distributions throughout the Holocene. Validations show that the CHELSA-TraCE21k V1.0 dataset reasonably represents the distribution of temperature and precipitation through time at an unprecedented 1 km spatial resolution, and simulations based on the data are capable of detecting known LGM refugia of species.

## 1 Introduction

Since the Last Glacial Maximum (LGM), variation in climate has caused multiple changes in the Earth surface, including the rearrangement of species distributions or even species extinctions (Prentice et al., 1991; Velichko et al., 1997; Adams and Faure, 1997; Williams et al., 2004; Yu et al., 2010; Binney et al., 2017). Yet we have not fully evaluated the historical underpinnings of these changes as we have often lacked the climate data at the necessary spatial resolution. Biological entities such as species usually encounter climatic conditions at spatial resolutions < 1 km (Seo et al., 2009) that are beyond the spatial resolution of numerical global circulation models (GCMs), which run at much coarser spatial resolution (e.g., > 0.5°). For many applications such as inference of ecological niches (Hutchinson, 1957), determination of growing seasons (McMaster and Wilhelm, 1997), identification of species migrations (Engler and Guisan, 2009), or modeling of high-resolution species distributions (Guisan and Zimmermann, 2000; Guisan and Thuiller, 2005), temporal and spatial variability in temperature and precipitation is of utmost importance. For such analyses, imprecisions in the underlying climate data can strongly deteriorate the analytical power (Soria-Auza et al., 2010).

For the recent past, the gap between the coarse GCM resolution and the high resolution needed for many ecological applications has been bridged using statistical downscaling (Wilby et al., 1998; Wood et al., 2004; Schmidli et al., 2006; Maraun et al., 2010; Karger et al., 2017a, 2020), dynamical downscaling (Skamarock et al., 2021), or interpolation of meteorological station data (Daly et al., 1997; Hijmans et

al., 2005; Meyer-Christoffer et al., 2015; Harris et al., 2020). While all of these methods work comparably well for current climatic conditions, station data are not available before the 19th century (end of the 20th century for satellite data), hampering an application of said methods to paleo-climatic models. Most paleo-climatic data at high spatial resolution are therefore based on climatologically aided interpolation (or the change factor method) of GCM output (Brown et al., 2018). This process uses the high-resolution information of current-day climatologies and adds an interpolated anomaly derived from a coarser-resolution GCM (Willmott and Robeson, 1995; Hunter and Meentemeyer, 2005). While this approach works rather well for short-term time series where topography is relatively stable (Daly et al., 1997), it becomes impractical for longer time series where the dependence structure between variables (e.g., topography and climate) is dynamic (Maraun, 2013). This phenomenon is of concern especially in the last 21 000 years, as the topography in many regions on Earth has changed drastically due to the retreating ice sheets and glaciers in polar regions and in high mountain areas (Scotese, 2001). While numerical climate models are able to simulate paleo-environmental conditions comparably well (Sepulchre et al., 2020), they are computationally very demanding, and therefore they have not been applied at ecologically relevant spatial resolutions of < 1 km yet. Current global kilometer-scale models only show a simulation throughput of 0.043 SYPD (simulated years per day) (Fuhrer et al., 2018), which is 25-fold lower than desired computationally efficient simulations of 1 SYPD (Schulthess et al., 2018; Schär et al., 2019). Even with state-of-the-art supercomputers and climate models this gap can only be minimized by a factor of 20 (Neumann et al., 2019).

Climate impact studies, however, often only use a reduced set of climate variables compared to those available from the output of numerical climate models (Frieler et al., 2017). Such studies therefore do not need a complete representation of all climate processes at high spatial resolution. In ecological studies, for instance, precipitation is often used along with minimum and maximum temperatures for analyses of species occurrences (Woodward et al., 1990). Also, it is common practice to describe species ranges by their climate envelopes; thus species distribution models (SDMs) are often built using a relatively small set of climate predictors based on monthly minimum and maximum temperature and precipitation (Guisan and Zimmermann, 2000; Guisan and Thuiller, 2005).

Here we present paleo-climatic data, downscaled from the CCSM3_TraCE21k (Transient climate evolution of the last 21 000 years using the Community Climate System Model Version 3) model output (hereafter: TraCE21k) to a 30 arcsec resolution using the CHELSA V1.2 (Climatologies at high resolution for the earth's land surface areas) algorithm (Karger et al., 2017a), which covers time steps of 100 years from 21 ka to 1950 plus four additional time steps until 1990

(TraCE21k), for minimum and maximum temperatures, surface precipitation, and paleo-orography.

## 2   Input data

### 2.1   TraCE-21k transient climate simulations

The TraCE-21k (Transient climate evolution of the last 21 000 years) simulation using the CCSM3 (Community Climate System Model Version 3) climate model (Liu et al., 2009; He, 2011; Marcott et al., 2011; Carlson et al., 2012) provides information on climate change over the last 21 000 years, i.e., from the Last Glacial Maximum (LGM, hereafter defined as 21 ka similar to Ehlers et al., 2011) to present. The TraCE-21k simulation reproduces many main features of post-glacial climate dynamics in various parts of the world from low to high latitudes and includes abrupt climate changes (Liu et al., 2009; He, 2011). The TraCE-21k simulation output has a T31_gx3v5 resolution (Otto-Bliesner et al., 2006). It uses a coarse-resolution dynamic global vegetation model (DGVM). The coupled atmosphere–ocean model in CCSM3 is based on the Community Atmospheric Model 3 (CAM3), on 26 vertical hybrid coordinate levels. The land and atmosphere components in CCSM3 in the TraCE-21k simulations use the same resolution. The parameterizations of the DGVM are largely based on the Lund–Potsdam–Jena (LPJ) DGVM. The ocean model in CCSM3 uses the NCAR (National Center for Atmospheric Research) version of the Parallel Ocean Program (POP) with 25 vertical levels, and the sea ice model is the NCAR Community Sea Ice Model (CSIM).

### 2.2   Observational climatology: CHELSA V1.2

CHELSA (Climatologies at high resolution for the earth's land surface areas) V1.2 is a high-resolution (30 arcsec) climate dataset for Earth's land surface areas (Karger et al., 2017b). It includes monthly means of daily 2 m mean, minimum, and maximum temperature and monthly precipitation rates at 30 arcsec resolution for the time period 1979–2013. CHELSA V1.2. is calculated with the CHELSA V1.2 topographic downscaling algorithm (Karger et al., 2017a), using the ERA-Interim (ECMWF Re-Analysis-Interim) reanalysis (Berrisford et al., 2009) as forcing data and GPCC (Global Precipitation Climatology Center) data (Meyer-Christoffer et al., 2015) for its bias correction.

### 2.3   Global model of glacial isostasy: ICE-6G_C (VM5a)

We used the output data of the ICE-6G_C (VM5a) (hereafter ICE6G) model as a basis for the extent of the major ice sheets at 1° resolution. ICE6G is a refinement of the ICE-5G (VM2) (hereafter ICE5G) global model of glacial isostasy (Peltier, 2004) which has been widely used to model the distribution of major ice sheets through time. ICE6G improves ICE5G

by applying all available global positioning system (GPS) measurements of vertical motion of the crust that constrain the thickness of local ice cover as well as the timing of its removal. ICE6G explicitly outputs changes in ice thickness of major ice sheets (e.g., the Laurentide ice sheet) from the LGM till today (Argus et al., 2014; Peltier et al., 2015) at 500-year time steps.

### 2.4 Observational glacial extent at Last Glacial Maximum (LGM)

As the extent of the glaciers during the LGM, we use data from Ehlers et al. (2011) that present a detailed overview of Quaternary glaciations all over the world, with regards not only to stratigraphy but also to major glacial landforms and the extent of the respective ice sheets.

### 2.5 Observational current glacial extent: GLIMS

The GLIMS (Global Land Ice Measurements from Space) project (Raup et al., 2007) at the NSIDC (National Snow and Ice Data Center) provides data on global glacial extent and other information about glaciers including metadata on how those outlines were derived. Here we use this database to delineate the current extent of the glaciers at high resolution globally.

### 2.6 Global Multi-resolution Terrain Elevation Data 2010 (GMTED2010)

The Global Multi-resolution Terrain Elevation Data 2010 (GMTED2010) (Danielson and Gesch, 2011) dataset contains elevation data for the globe collected from various sources. Here we use the 30 arcsec version of the data that represents the mean elevation of all 7.5 arcsec grid cells that represent the highest available resolution of the data.

### 2.7 Bathymetric DEM

We use the General Bathymetric Chart of the Oceans (GEBCO) 2014 (Weatherall et al., 2015) as bathymetry. Although GEBCO also includes land surface altitude, we only use it for the oceans, and we keep as land altimetric data those of the CHELSA V1.2 algorithm (that being GMTED2010) to maintain comparable topography at the land surface.

### 2.8 Global sea level change

We used data from Miller et al. (2005) for the estimation of global sea level change from 21 ka to 1990. The data provide global estimates of sea level change over the last 100 million years. The entire time series of sea level change is based on a variety of proxy data, with the data used here dating back to the LGM, mainly based on tropical reef proxies (Miller et al., 2005).

## 3 Methods

Downscaling is based on the CHELSA V1.2 algorithm (Karger et al., 2017a) using forcing from TraCE-21k simulations (Liu et al., 2009; He, 2011) and involving several processing steps (Fig. 1). The CHELSA V1.2 algorithm needs a dynamic forcing in the form of GCM output (Karger et al., 2020) or gridded reanalysis data (Karger et al., 2017a, 2021b), as well as a surface orography (i.e., topography above sea level) to run a suite of downscaling algorithms for key climatic variables such as air temperature and precipitation. As the orography at different time steps between 21 ka and current times is not available at the high resolution required for the CHELSA V1.2 algorithm, we approximated it using a combination of data from the digital elevation model GMTED2010 (Danielson and Gesch, 2011), large-scale ice sheet configurations from ICE6G (Peltier et al., 2015), high-resolution glacier extents from GLIMS for current conditions (Raup et al., 2007) and LGM conditions (Ehlers et al., 2011), and sea level change data from Miller et al. (2005) (Fig. 1). We then ran the CHELSA V1.2 algorithm on the paleo-orography using a bias-corrected version of the TraCE-21k simulations as a forcing. Details on these steps are described in the following sub-sections.

### 3.1 Paleo-orography

The first step in estimating the paleo-orography was carried out for the LGM (21 ka). For this time point, both estimates of glacial extents from Ehlers et al. (2011) and estimates of glacier thickness from ICE6G exist. We first combined the topographic information from GMTED2010 on land and that of GEBCO into a bedrock topography that provides the current bedrock topography $e_c^{topo}$ (including current-day glaciers; see ff.). To create a bedrock orography $e_t^{bed}$ (i.e., topography adjusted for sea level without glaciers except for currently glaciated areas), we used the information on past sea level changes and set all elevations to 0 so that

$$e_t^{bed} = e_c^{topo} - sl_t. \tag{1}$$

To include the orography of the glaciers we first converted the polygons of the glacial extents from Ehlers et al. (2011) into point locations (Fig. 2a, black dots) and extracted their elevation from $e_t^{bed}$ (Fig. 2a) at time $t = 0$ (LGM, 21 ka), resulting in the surface elevation of the glacial boundaries (gb) $e_{gb_t}^{bed}$. To combine the high-resolution estimates from Ehlers et al. (2011), with the coarser (1°) resolution of ICE6G, we randomly sampled 100 point locations per 1° grid cell from ICE6G and again extracted the surface elevation of the glaciers from $e_t^{bed}$ at the ICE6G time step that is nearest to the time step $t = 0$ (LGM, 21 ka), resulting in $e_{G_t}$ (Fig. 2b). All points that did not fall within the high-resolution glacial extent were omitted so that only points within the high-resolution estimate of glacial extent from Ehlers et al. (2011) remained (Fig. 2b). Then both point datasets $e_{gb_t}^{bed}$ and $e_{ICE6G_t}^{orog}$

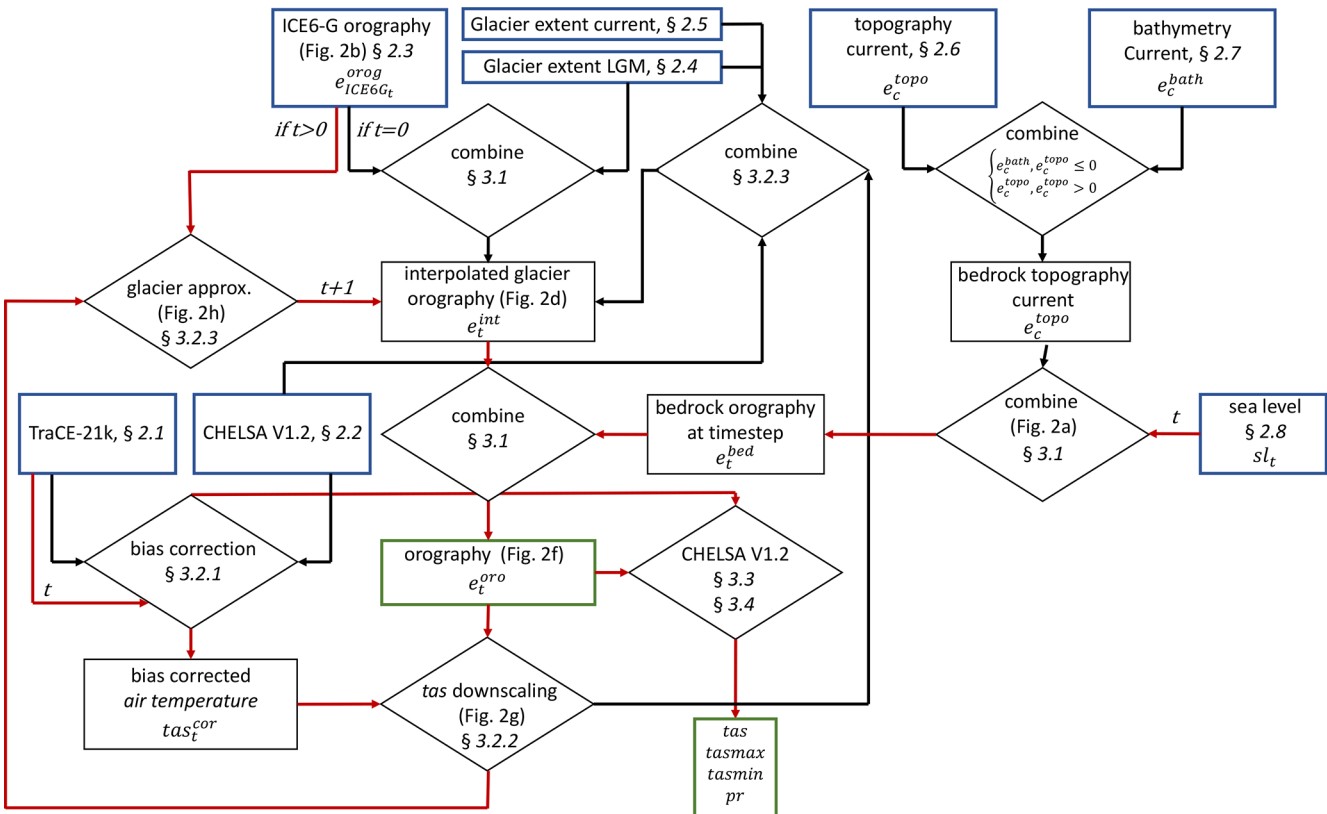

**Figure 1.** Graphical representation of the different steps employed in downscaling TraCE-21k simulations using the CHELSA V1.2 algorithm. Input datasets are indicated by a blue box; output data are indicated by a green box. Rhombi indicate processing steps; $t$ indicates discrete time steps, with $t = 0$ being the LGM. Red lines indicate processing steps that are run iteratively over all time steps; black lines indicate computations that were run only once.

were combined to represent a point sample of the surface elevation $e_{G_t}$ within the high-resolution glacial extent of Ehlers et al. (2011) (Fig. 2c). Next, this point sample was spatially interpolated to a grid of 30 arcsec resolution applying multilevel $B$-spline interpolations. By this, we achieved an interpolated gap-free high-resolution estimate of glacial surface elevation $e_t^{int}$ at $t = 0$ (LGM, 21 ka) (Fig. 2d). The multilevel $B$-splines use a $B$-spline approximation to $e_{G_t}$ and start using the coarsest grid $\phi_0$ from an overall set of grids $\phi_0, \phi_1, \ldots, \phi_n$, with $n = 14$ generated using optimized $B$-spline refinements (Lee et al., 1997). The resulting $B$-spline function $f_0(e_{G_t})$ then gives the first approximation of $e_t^{int} = f_0(e_{G_t})$ and leaves a deviation

$$\Delta^1 e_t^{int} = e_t^{int} - f_0(x_c, y_c) \tag{2}$$

at each grid cell $c$ location $\left(x_c, y_c, e_t^{int}\right)$. Then the next control lattice $\phi_1$ is used to approximate $f_1(\Delta^1 e_t^{int}{}_c)$. This approximation is then repeated on the sum of

$$f_0 + f_1 = e_t^{int} - f_0(x_c, y_c) - f_1(x_c, y_c) \tag{3}$$

at each grid cell $c$ $\left(x_c, y_c, e_t^{int}{}_c\right) n$ times, resulting, in our case, in the gap-free interpolated glacial surface $e_t^{int}$. The interpo-

lated glacial surface was then combined with $e_t^{bed}$ to the topography $e_t$ (Fig. 2e) using

$$e_t = \begin{cases} e_t^{topo}, & e_t^{topo} \geq e_t^{int} \\ e_t^{int}, & \text{otherwise}. \end{cases} \tag{4}$$

The final orography $e_t^{oro}$ at time step $t = 0$ (i.e., topography above sea level) (Fig. 2f) is then generated using

$$e_t^{oro} = \begin{cases} 0, & e_t \leq \text{sl}_t \\ e_t, & e_t > \text{sl}_t, \end{cases} \tag{5}$$

with $\text{sl}_t$ being the sea level at time step $t$. Although this approach includes changes in the glacial surface and sea level rise, it ignores changes in bedrock elevation due to upwelling after glacier melt.

## 3.2 Interpolation of glacier extent and thickness between LGM and current

As high-resolution estimates of glacial surface elevation are not available for time steps $t$ other than the LGM and current day, we use a combination of mean annual 2 m air temperature data together with sea surface elevation $\text{sl}_t$ and

ICE6G orography $e_{\text{ICE6G}_t}^{\text{orog}}$ data to estimate $e_t^{\text{oro}}$ at each time step $t \neq 0$ and $t = 221$. The rationale behind this approach is that temperature and glacier extents are interdependent, and a change in temperature will translate into a change in glacial extent or thickness. The procedure to generate high-resolution glacial surfaces is explained in the following sections.

### 3.2.1 Bias correction of air temperature

In a first step, the orography $e_t^{\text{oro}}$ with $t = 0$ (LGM, 21 ka) was used to downscale mean annual temperature $\text{tas}_t$. GCMs such as the CCSM3 normally exhibit a large bias in temperatures or precipitation (Cannon et al., 2015; Maraun, 2013). We therefore applied a change factor bias correction based on the bias observed between current annual-mean 2 m air temperatures $\text{tas}_{\text{cur}}^{\text{obs}}$ from CHELSA V1.2 normals resampled to a 0.5° grid resolution and that of TraCE21k simulated for the same time period $\text{tas}_{\text{cur}}^{\text{mod}}$, spline-interpolated using the same multilevel $B$-spline interpolation method as described in Sect. 3.1 to 0.5° grid resolution. The resolution of 0.5° follows the same procedure as used in the CHELSA V1.2 algorithm (Karger et al., 2017a). We used the time period 1980–1990 ($= \text{cur}$) to calculate this bias, as it is the only time period for which CHELSA V1.2 data and TraCE21k overlap. The change factor was then calculated as

$$\Delta\text{tas} = \text{tas}_{\text{cur}}^{\text{obs}} - \text{tas}_{\text{cur}}^{\text{mod}}. \tag{6}$$

This effectively preserves the trends observed in temperature but simultaneously assumes that the bias has also been conserved over time (Maraun, 2016). The bias-corrected temperatures $\text{tas}_t^{\text{cor}}$ are then given by

$$\text{tas}_t^{\text{cor}} = \text{tas}_{\text{cur}}^{\text{obs}} - \Delta\text{tas}. \tag{7}$$

### 3.2.2 Downscaling mean annual air temperature

To achieve a high-resolution approximation of near-surface air temperatures (Fig. 2g), we used a lapse-rate-based downscaling from atmospheric temperature data at the TraCE-21k pressure levels. The lapse rates $\Gamma$ are based on a linear approximation from average temperatures $\text{ta}_z$ at altitudes $a_z$ and vertical levels 26 (992.5 hPa) to 20 (600.5 hPa) of the T31_gx3v5 grid that contain all surface elevations so that

$$\Gamma = \frac{n(\sum_{a_z \text{ta}_z}) - (\sum_{a_z})(\sum_{\text{ta}_z})}{n(\sum_{a_z^2}) - (\sum_{a_z})^2}. \tag{8}$$

Temperature at the surface at a high spatial resolution (tas) was then calculated by

$$\text{tas}_t = \Gamma_t \cdot e_t^{\text{oro}} + \text{tas}_t^{\text{cor}}. \tag{9}$$

### 3.2.3 Glacier extent approximation using mean annual air temperature

We assume that air temperature is correlated to glacier extent and use this relationship to estimate the boundaries of glaciers for each time step separately. To do so, we use mean annual 2 m air temperature at the boundary of the interpolated high-resolution glacier orography. We then transformed the glacier elevations $e_t^{\text{int}}$ to a polygon $G_t$ and then transformed the outline of this polygon to a point sampling of the glacier boundaries at time $t = 0$. Mean annual 2 m air temperatures at this glacier boundary were then extracted for this point sample, which gives the local annual-mean air temperature $\text{tas}_{t=0}^{\text{gb}}$ under which a glacier had a boundary at the LGM. To set this in relation to current mean annual air temperatures at current glacier boundaries $\text{tas}_{t=221}^{\text{gb}}$ (with $t = 221$ being the year 1990), we calculated the difference between current and LGM boundary temperatures. The resulting point locations for both $\text{tas}_{t=0}^{\text{gb}}$ and $\text{tas}_{t=221}^{\text{gb}}$ were then spatially interpolated using a multilevel $B$-spline (as described in Sect. 3.1) to result in a gap-free surface and then subtracted, resulting in $\Delta\text{tas}_{\text{cur}}^{\text{obs}}$ (Fig. 2g).

As the orography for the next time step is not known yet, we estimated the near-surface air temperature $\text{tas}_{t+1}^{\text{est}}$ for the reduction in glacier extent similarly to the time step before so that

$$\text{tas}_{t+1}^{\text{est}} = \Gamma_{t+1} * e_t^{\text{oro}} + \text{tas}_{t+1}^{\text{cor}}. \tag{10}$$

The binary glacial extent $G_{t+1}$ at $t + 1$ is then approximated as

$$G_{t+1} = \begin{cases} 1, \text{tas}_{t+1}^{\text{est}} < \text{tas}_t^{\text{gb}} + \frac{\Delta\text{tas}_{\text{cur}}^{\text{obs}}}{-1 \cdot (\sum t - t)} \wedge G_0 = 1 \\ 0, \text{otherwise}. \end{cases} \tag{11}$$

This glacial extent is then used again to estimate the combined topography at $t + 1$ in the same way as described in Sect. 3.1. As ICE6G has a 500-year resolution we used the ICE6G orography that is closest to each time step. As ICE6G only includes information on the major ice sheets, smaller ice sheets in the Alps do not include a sample of $e_{\text{ICE6G}_t}^{\text{orog}}$. In the case of smaller ice sheets, the surface orography from ICE6G is replaced by a point sample of the elevation of the glacier boundary under current conditions $e_{\text{gb}_c}^{\text{orog}}$ (Fig. 2h). The glacier orography in this case is then created by using a spline interpolation between $e_{\text{gb}_t}^{\text{bed}}$ and $e_{\text{gb}_c}^{\text{orog}}$. In Eq. (11) the second term in the condition for $G_{t+1}$ linearly scales $\Delta\text{tas}_{\text{cur}}^{\text{obs}}$ over the entire number of time steps. This correction is necessary, as otherwise the entire bias would be added at the first time step, resulting in an unrealistically strong shift in the glacial extent. We then repeated the transformation of the glacial extent $G_{t+1}$ to all point locations and repeated the procedure for the temperature coupling to estimate the orography $e_{t+1}^{\text{oro}}$. Near-surface air temperatures for $t + 1$ have then been approximated using

$$\text{tas}_{t+1} = \Gamma_{t+1} \cdot e_{t+1}^{\text{oro}} + \text{tas}_{t+1}^{\text{cor}}. \tag{12}$$

## 3.3 Downscaling mean monthly precipitation rates

### 3.3.1 Orographic wind effects

The estimation of high-resolution precipitation follows a variant of the CHELSA V1.2 algorithm (Karger et al., 2017a). The CHELSA V1.2 algorithm assumes that orography is one of the main drivers of precipitation (Spreen, 1947; Basist et al., 1994; Daly et al., 1997; Sevruk, 1997; Böhner, 2006; Gao et al., 2006; Böhner and Antonic, 2009; Karger et al., 2017a). In tropical convective regimes, precipitation typically increases up to the condensation level around 1000–1500 m above surface, while the exponentially decreasing moisture content in the mid- to upper troposphere results in a drying above the condensation level and in non-linear precipitation lapse rates (Körner, 2007). Furthermore, negative precipitation lapse rates are common under the extremely dry polar climates. In contrast, at mid-latitudes and in the subtropics, precipitation generally increases with increasing elevation due to advection. As a consequence, summits of the Alps or other high mountain ranges exhibit high rainfall (Rotunno and Houze, 2007), and lapse rates for precipitation are almost linear (Weischet and Endlicher, 2008). To approximate the effects of orographic precipitation we used the CHELSA V1.2 algorithm, which is explained in more detail below.

We used 10 m $u$-wind and $v$-wind components of TraCE-21k to calculate wind direction. Both wind components were projected to a world Mercator projection at a 4 km grid resolution using a multilevel $B$-spline interpolation similar to the one described in Sect. 3.1. Windward and leeward effects are assumed to be best represented at resolutions larger than 1 km (Daly et al., 1994); we therefore chose a grid resolution of 4 km for the underlying digital elevation model. The wind effect $H$ was then calculated using

$$H_{\mathrm{W}} = \frac{\sum_{i=1}^{n} \frac{1}{d_{\mathrm{WHi}}} \tan^{-1}\left(\frac{d_{\mathrm{WZi}}}{d_{\mathrm{WHi}}^{0.5}}\right)}{\sum_{i=1}^{n} \frac{1}{d_{\mathrm{LHi}}}} + \frac{\sum_{i=1}^{n} \frac{1}{d_{\mathrm{LHi}}} \tan^{-1}\left(\frac{d_{\mathrm{LZi}}}{d_{\mathrm{LHi}}^{0.5}}\right)}{\sum_{i=1}^{n} \frac{1}{d_{\mathrm{LHi}}}} \quad (13)$$

$$H_{\mathrm{L}} \frac{\sum_{i=1}^{n} \frac{1}{\ln(d_{\mathrm{WHi}})} \tan^{-1}\left(\frac{d_{\mathrm{LZi}}}{d_{\mathrm{WHi}}^{0.5}}\right)}{\sum_{i=1}^{n} \frac{1}{\ln(d_{\mathrm{LHi}})}}, \quad (14)$$

where $d_{\mathrm{WHi}}$ and $d_{\mathrm{LHi}}$ refer to the horizontal distances between the focal 4 km grid cell in the windward and leeward direction, and $d_{\mathrm{WZi}}$ and $d_{\mathrm{LZi}}$ are the corresponding vertical distances compared with the focal 4 km cell following the wind trajectory. The second summand in the equation for $H_{\mathrm{W}}$, where $d_{\mathrm{LHi}} < 0$, accounts for the leeward impact of previously traversed mountain chains. The horizontal distances in the equation for $H_{\mathrm{L}}$, where $d_{\mathrm{LHi}} \geq 0$, lead to a longer-distance impact of leeward rain shadow. The final wind effect parameter is calculated as $H = H_{\mathrm{L}} H_{\mathrm{W}}$. Both equations were applied to each grid cell at the 30 arcsec resolution in a world Mercator projection. Orographic precipitation effects

are less pronounced just above the surface, as well as in the free atmosphere above the planetary boundary layer (Daly et al., 1997; Oke, 2002; Stull, 1988; Karger et al., 2020). The highest impact of orography is considered just at the boundary layer height where the airflow interacts with the terrain. We used the lifted condensation level (LCL) as an indicator of the altitude at which the wind effect exerts the highest contribution to precipitation. The LCL has been calculated using the mean air temperature (tas) and mean near-surface relative humidity (hurs) using

$$\mathrm{LCL} = 20 + (\mathrm{tas}/5) \cdot (100 - \mathrm{hurs}) \quad (15)$$

(Lawrence, 2005). The LCL has been interpolated to a 30 arcsec resolution using a $B$-spline interpolation. To create a boundary-layer-height-corrected wind effect $H_{\mathrm{B}}$, the wind effect grid $H$ containing LCL was then proportionally distributed to all grid cells falling within a respective T31 grid cell using

$$H_{\mathrm{B}} = \frac{H}{1 - \left(\frac{|z - \mathrm{LCL}_z| - z_{\max}}{h}\right)}, \quad (16)$$

with $z_{\max}$ being the maximum distance between the LCL at elevation $z$ and all grid cells at a 30 arcsec resolution falling within a respective T31 grid cell. In Eq. (16), $h$ is a constant of 9000 m, and $z$ is the respective elevation from GMTED2010 (Danielson and Gesch, 2011) with

$$\mathrm{LCL}_z = \mathrm{LCL} + z_{\mathrm{GCM}} + f, \quad (17)$$

$z_{\mathrm{GCM}}$ being the elevation of the TraCE21k grid cell and $f$ being a constant of 500 m which takes into account that the level of highest precipitation is not necessarily at the lower bound of the LCL, but slightly higher (Karger et al., 2017a).

The wind effect algorithm cannot distinguish extremely isolated valleys inside highly elevated mountain areas (Frei and Schär, 1998). Such valleys are situated in areas where the wet air masses flow over an orographic barrier and are prevented from flowing into deep valleys. These effects are mainly confined to large mountain ranges and are not as prominent in small- to intermediate-sized mountain ranges (Liu et al., 2013). To account for these effects, we used a variant of the windward–leeward equations with a linear search distance of 300 km in circular steps of 5° from 0 to 355° for each grid cell. The calculated leeward index was then scaled towards higher elevations using

$$E = \left(\frac{\sum_{i=1}^{n} \frac{1}{d_{\mathrm{WHi}}} \tan^{-1}\left(\frac{d_{\mathrm{LZi}}}{d_{\mathrm{WHi}}}\right)}{\sum_{i=1}^{n} \frac{1}{d_{\mathrm{LHi}}}}\right)^{\frac{z}{h}}. \quad (18)$$

~~The $e$ value was set to 9000 m, and~~ $h$ has been set to 9000 m. $E \cdot H_{\mathrm{B}}$ will give the first approximation of the orographic precipitation intensity $p_{\mathrm{I}}$.

### 3.3.2 Bias-correcting precipitation and downscaling

Precipitation, similar to temperature, exhibits a rather large bias in TraCE-21k (Figs. 3, 4). To remove this bias, we applied a change factor bias correction similar to the one described in Sect. 3.2.1 with the reference period 1980–1990. Here, we used a multiplicative change factor to avoid precipitation rates $< 0$. Additionally, we included a constant of $c = 0.0001 \, \mathrm{kg \, m^{-2}}$ per month to avoid division by zero so that

$$\Delta \mathrm{pr_m} = (\mathrm{pr_{cur_m}^{mod}} + c)/(\mathrm{pr_{cur_m}^{obs}} + c), \qquad (19)$$

with m being the respective month of the year. The bias-corrected precipitation rate for $p_m^{cor}$ is then calculated by

$$\mathrm{pr_m^{cor}} = \mathrm{pr_{cur_m}^{obs}}/\Delta \mathrm{pr_m}. \qquad (20)$$

To achieve the distribution of monthly precipitation $\mathrm{pr_o}$ given the approximated orographic precipitation intensity $p_{I_c}$ at each grid location $(x_c, y_c)$, we used a linear relationship between $\mathrm{pr_m^{cor}}$ and $\mathrm{pr}_{I_c}$ using

$$\mathrm{pr}_o = \frac{p_{I_c}}{\frac{1}{n}\sum_{i=1}^{n} p_{I_{c_i}}} \cdot \mathrm{pr_m^{cor}}, \qquad (21)$$

where $n$ equals the number of 30 arcsec grid cells of $p_I$ that fall within a 0.5 grid cell of $p_m^{cor}$.

### 3.4 Downscaling mean monthly near-surface air temperatures

The downscaling of monthly near-surface air temperatures (tas, tasmax, tasmin) follows the methods described in Sect. 3.2.2, with the only difference being that instead of mean annual temperature, tasmax and tasmin are used, where $\mathrm{tas} = (\mathrm{tasmax} + \mathrm{tasmin})/2$. The temperatures have again first been bias-corrected using

$$\Delta \mathrm{tasmax_m} = \mathrm{tasmax_{cur_m}^{obs}} - \mathrm{tasmax_{cur_m}^{mod}} \qquad (22)$$

$$\Delta \mathrm{tasmin_m} = \mathrm{tasmin_{cur_m}^{obs}} - \mathrm{tasmin_{cur_m}^{mod}} \qquad (23)$$

and

$$\mathrm{tasmax_{m_t}^{cor}} = \mathrm{tasmax_{cur_m}^{obs}} - \Delta \mathrm{tasmax_m} \qquad (24)$$

$$\mathrm{tasmin_{m_t}^{cor}} = \mathrm{tasmin_{cur_m}^{obs}} - \Delta \mathrm{tasmin_m}, \qquad (25)$$

with m being the respective month of the year in Eqs. (22)–(25).

## 4 Output validation

Direct validation of the temperature (Fig. 1) and precipitation (Fig. 2) output at high resolution for paleo-time series relies on proxies, as direct observations of both variables are not available. Although global temperature time series exist,

they only give global means and do not allow validation of the performance of a 1 km paleo-climatic dataset. Therefore, to validate the CHELSA-TraCE21k dataset we complement a simple comparison of the simulated time series to proxy data and current observations with approaches of validating derived parameters from the simulated temperature and precipitation that directly benefit from a very high horizontal resolution.

### 4.1 Validation using current (historical) observations

We used data from the Global Historical Climate Network (GHCN) monthly database V.3 (Lawrimore et al., 2011) to validate the performance of the downscaling algorithm during the last time step of the CHELSA-TraCE21k for which station data are available. To do so we calculated monthly climatologies for each month for tasmax, tasmin, and pr from both TraCE-21k and CHELSA-TraCE21k. We then compared the values measured at each station to those simulated in both TraCE-21k and CHELSA-TraCE21k.

The original TraCE-21k data show large deviations and root mean square errors (RMSEs) from the observed data (Fig. 3). This is expected as a climate model running for such long time periods needs to have coarse resolution, as well as a large degree of generality and realism, which decreases the accuracy of a model when compared to observations. The temperature variables perform well in TraCE-21k, with $r \sim 0.8$ for all months, but have deviations and RMSE similar to those of precipitation, which most likely can be attributed to the coarse resolution of the climate model. TraCE-21k also seems to overestimate temperature extremes for both tasmax and tasmin (Fig. 4).

The precipitation, however, does not perform well in the model, with $r \sim 0.4$ and large deviations from actual values (Fig. 3), and overall precipitation seems to be too low in the model (Fig. 4).

The CHELSA V1.2 algorithm improves the correlation between observed and modeled data and decreases the standard deviation for all three parameters (Fig. 3). The downscaling for the temperature variables increases the correlation to $r \sim 0.95$ for all months and decreases the standard deviation substantially (Fig. 3). Similarly, the performance of the precipitation estimation in CHELSA-TraCE21k increases, which is reflected in an $r$ value of $\sim 0.7$ and a lower standard deviation and RMSE (Fig. 3). The underestimation of precipitation in the TraCE21k is reduced, but the downscaling algorithm still has a considerable bias (Fig. 3) during the historical period.

### 4.2 Comparison with temperature proxies from ice core data

We compared the downscaled temperatures with the Greenland ice core reconstructions of Buizert et al. (2014, 2018) to check the performance of the downscaling at eight ice

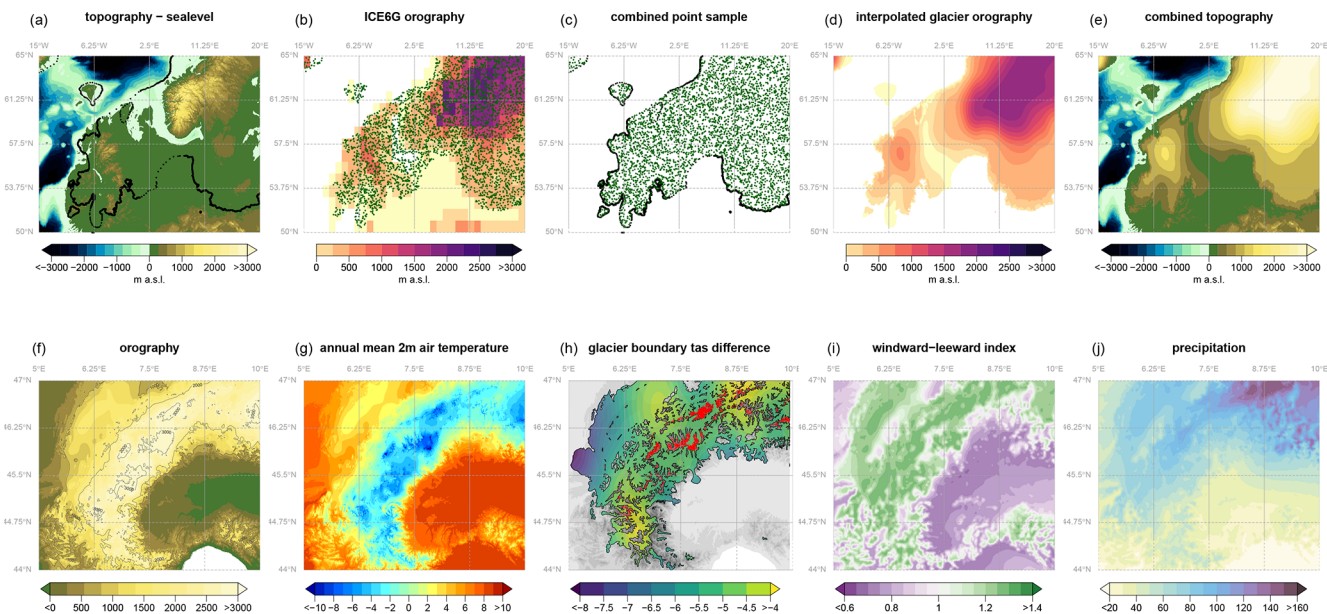

**Figure 2.** Illustration of several steps performed to estimate the surface orography and the temperature and precipitation fields in CHELSA-TraCE21k during the Last Glacial Maximum (21 ka). The upper row gives an example of the interpolation of the European ice sheets; the lower row shows an example of the resulting orography and environmental variables in the western part of the European Alps. **(a)** Topographic information at $t = 0$ (LGM) is combined with information on past sea levels and the boundary of ice sheets (black dots) for which the surface elevation is extracted. **(b)** Within the extent of ice sheets, surface elevation is extracted from the ICE6G orography for a random sample of points for $t = 0$. **(c)** Both point samples from **(a)** and **(b)** are combined and interpolated **(d)** to estimate the orography of the glaciers. **(e)** The interpolated glacier orography and the sea-level-adjusted topography are then combined. **(f)** The high-resolution (30 arcsec) orography (shown here for the western Alps) is then used as a basis at $t = 0$ for **(g)** a lapse-rate-based downscaling of air temperature. **(h)** From the high-resolution temperatures, information on the glacier boundaries during the LGM (black) and current times (red) is extracted, and the difference is interpolated to correct the temperature-based shrinking and expansion of the glaciers. **(i)** Based on the orography the windward–leeward index is calculated (shown for July 21 ka), which builds the basis for the **(j)** precipitation approximation.

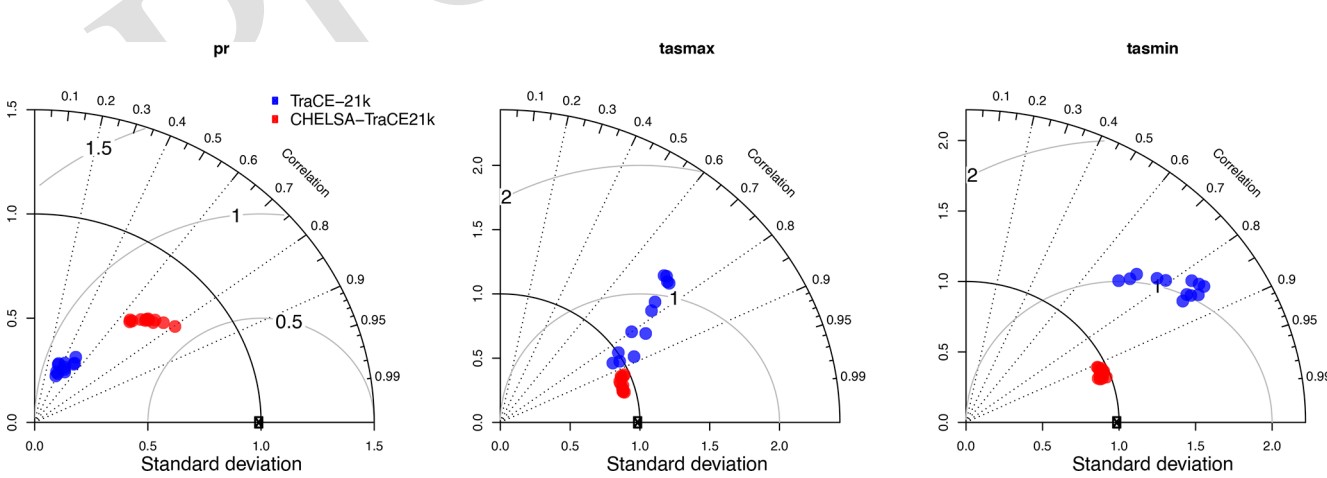

**Figure 3.** Taylor diagrams comparing the relationship between TraCE-21k (blue) and CHELSA-TraCE21k (red). Data are shown for the 20th-century time period with average monthly observational data from the Global Historical Climate Network (GHCN) for the time period 1950–1990. Each dot represents a specific month.

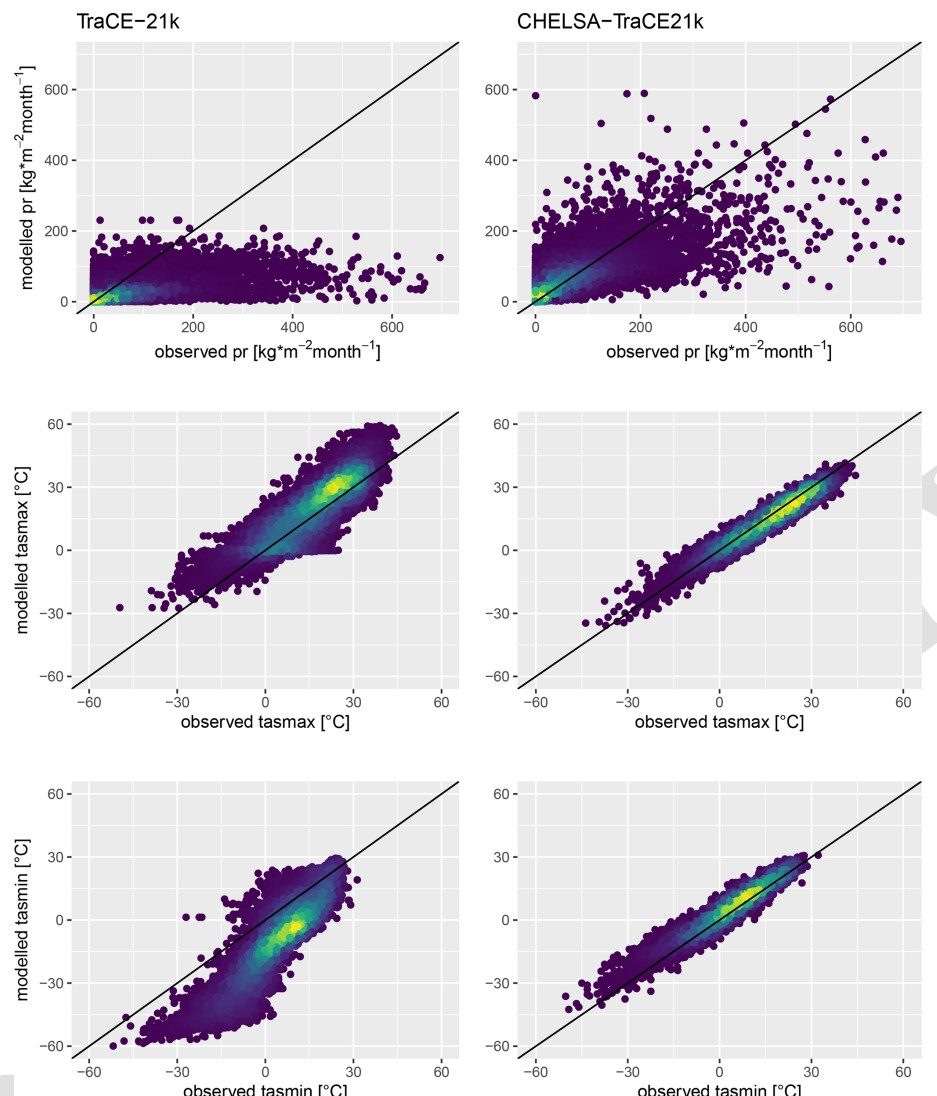

**Figure 4.** Scatterplots comparing precipitation, maximum, and minimum temperature. Data are aggregated from TraCE-21k and CHELSA-TraCE21k for the 20th-century time period with observational data from the Global Historical Climate Network (GHCN) for the time period 1950–1990.

core locations on the Greenland Ice Sheet (GIS). Although both temperature reconstructions and GCM-generated temperatures have uncertainties connected to them (Erb et al., 2018), the ice core data are so far the best possible validation dataset that spans the entire deglaciation period from 21 ka to 1990 (Buizert et al., 2014, 2018). To assess the performance gain of the downscaling over the coarse-resolution TraCE21k data, we compare the ice core annual-mean near-surface temperature reconstructions with both the CHELSA-TraCE21k and the original TraCE-21k temperature data.

Compared to the temperature reconstructions from ice cores, the downscaled CHELSA-TraCE21k model had reduced bias at four of the ice core sites located at the edges of the GIS (ReCAP, Agassiz, Hans Tausen Iskappe, Camp Century) but increased the bias, RMSE, and mean absolute

error (MAE) at the remaining four sites at the center of the GIS (NEEM, NGRIP, GISP2, Dye 3) (Fig. 5). Overall, both CHELSA-TraCE21k and TraCE21k show a warm bias before the Heinrich 1 event (i.e., the break-off of large groups of icebergs from Greenland into the North Atlantic, 16.8 ka) and roughly after the 8.2 ka event at four of the sites (ReCAP, Agassiz, Hans Tausen Iskappe, Camp Century). At four sites (ReCAP, Agassiz, Hans Tausen Iskappe, Camp Century) a cold bias is present after the Younger Dryas (Fig. 5). At the four other sites, CHELSA-TraCE21k usually shows a warm bias before the H1 and TraCE-21k a cold bias before the H1 (Fig. 5). After the Younger Dryas (12.9–11.6 ka), both models show a cold bias at these sites. At the Camp Century site, the TraCE-21k data are close to the $\delta^{15}$N-based temperature reconstructions before the H1 event, and CHELSA-

TraCE21k shows a warm bias, while after the Younger Dryas the situation is reversed (Fig. 5).

The bias observed after downscaling might be related to biases in all the different input sources, such as the TraCE-21k bias being amplified, a bias in the ice core proxy data itself, or the bias correction using the simple change factor method. With the available data, these potential causes cannot be clearly disentangled but should be kept in mind for applications of the data.

## 4.3 Validation of glacier extent

Although the downscaling algorithm might increase the performance of the temperature and precipitation estimates during the historical period, this does not imply that this improvement is equal during the entire transient time series. To further validate the data, we therefore compared it to more derived parameters for which time series data exist.

As the ice core temperature reconstructions have associated uncertainties, it is impossible to disentangle if potential differences between the ice core data and the model data are due to uncertainties in the reconstructions. To validate the downscaled temperature data further, we used the interpolated extent of glaciers in CHELSA-TraCE21k and compared it to glacial-extent data from Dyke (2004). The data consist of expertly delineated glacier maps based on a chronological database of radiocarbon dates and contain > 4000 dates located in North America (Dyke, 2004). To compare both datasets, we first calculated the glacial extent from CHELSA-TraCE21k by assigning a binary value to each 1 km grid cell in a Lambert conformal conic projection so that each data point compared equals 1 km$^2$ either being covered by a glacier [1] or being free of a glacier [0]. We assigned a 1 if the simulated glacier height was above the paleo-terrain elevation and a 0 if it was lower than or equal to the paleo-terrain elevation. The paleo-terrain elevation was calculated using the current terrain elevation minus the sea level difference between current day and that of the respective paleo-time step. As the current terrain elevation already includes extent glaciers, this elevation-dependent procedure of assigning glacial extents would result in the current glaciers being assigned a 0. Therefore, we assigned all grid cells covered by extent glaciers a 1.

To compare the simulated glacial extent to the expertly delineated extent, we rasterized the polygons provided by Dyke (2004) for the years 18–1 ka to the 1 km resolution, extent, and projection of the simulated glacial cover and assign a 1 where the polygon intersects with a 1 km raster cell and a 0 otherwise.

We then calculated three different test values to identify if the simulations correctly predict the presence and absence of a glacier. As the dataset is highly unbalanced between absences of glaciers [0] and presences of glaciers [1] through time we use balanced accuracy, which is defined as (sensitivity + specificity)/2. Additionally we report Cohen's kappa and the true-skill statistic (Allouche et al., 2006).

The test validations of the glacial extent show a good performance over most time steps (Fig. 6), but with a notable drop in accuracy at 8 ka, where all validation metrics drop significantly. Aside from the drop at 8 ka, the performance of the glacial-extent simulations performed well. The marked drop in performance around 8 ka might be due to the 8.2 ka event, which marked a strong decrease in global temperatures, most likely due to meltwater fluxes from the collapsing Laurentide ice sheet. The strong coupling between temperature and glacial extent in CHELSA-TraCE21k generates an increase in glacial extent more than a sudden collapse during this time period, which seems to override the signal from the ICE6G forcing data in CHELSA-TraCE21k. Additionally, we used the data from the extent of the ice sheets over Fennoscandia from 22 to 10 ka (Stroeven et al., 2016) for all time steps for which ICE6G data and data from Stroeven et al. (2016) were available. The results (Supplement Fig. S1) show, similar to the Laurentide ice sheet, that the accuracy is relatively high until 10.5 ka, with a drop in accuracy at 10 ka. Therefore, we assume that the temperature coupling does introduce errors in the time between 10 and 6 ka, as is evident from the comparison with the ice sheets of North America and over Fennoscandia.

## 5 Plausibility test using dynamic simulation of effective plant refugia

Transient long-term climatic data have a wide range of possible applications, ranging from population genetics (Leugger et al., 2022; Yannic et al., 2020), community ecology (Staples et al., 2022), and biodiversity buildup (Garcés-Pastor et al., 2022; Alsos et al., 2022) to evolutionary biology (Cerezer et al., 2022), just to name a few. Here we use one application in paleoecology as a plausibility test to additionally check if the transient CHELSA-TraCE21k data can reliably detect known LGM refugia of plant species. Climatic changes during the last glacial cycle since the LGM have had a significant influence on the distribution of ecosystems (Williams and Jackson, 2007), species (Hewitt, 1999; Hampe and Jump, 2011), and as a result on intraspecific genetic structures and speciation (Alsos et al., 2012; Yannic et al., 2014, 2020; Pellissier et al., 2015).

Tracing the distribution of species through time is, however, challenging as the spatio-temporal distributions of species strongly depend on environmental suitability (Guisan and Zimmermann, 2000), spatial accessibility of a given location (Svenning and Skov, 2004; Normand et al., 2011), and species dispersal abilities (Engler and Guisan, 2009). A dynamic simulation of species distributions can integrate all these aspects and therefore provides a valuable test bed for climatic data (Nobis and Normand, 2014). However, the spatio-temporal resolution of climate data needed for such

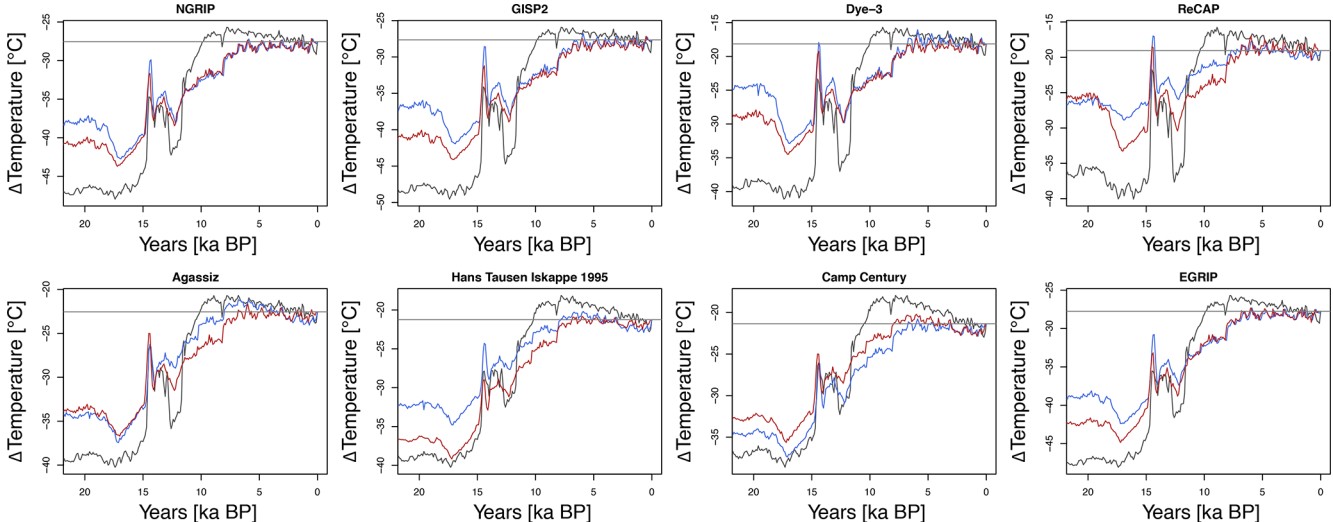

**Figure 5.** Comparison of temperature anomalies from current times (1950–1990) for the CHELSA-TraCE21k time series data (blue), the TraCE21k data (red), and the temperature reconstructions from ice cores (black) for eight sites across Greenland. The gray horizontal line indicates the current observed temperature during the period 1981–2010 from CHELSA V2.1. data. Temperatures are plotted as anomalies from the current temperature recorded at the respective location of the ice cores.

simulations has been limited to comparable coarse-grain climatic data (Gherghel and Martin, 2020), which usually creates a mismatch between the climate derived from the model and the climate actually experienced by an organism (Seo et al., 2009).

Here, we use the downscaled transient temperature and precipitation from CHELSA-TraCE21 since 17 ka (the coldest recorded temperatures in the CHELSA-TraCE21k model for Europe) to reconstruct refugia of the deciduous tree gray alder (*Alnus incana*) in Europe before post-glacial climate warming. Similar to Nobis and Normand (2014), we first calibrated a generalized linear model (GLM) (Nelder and Wedderburn, 1972) using current presences and absences of gray alder within polygons of the Atlas Florae Europaeae (AFE) (Jalas and Suominen, 1976) as the response variable and current annual-mean temperature and precipitation from CHELSA-TraCE21k as predictors calculated as zonal-mean values of $5 \times 5$ km rasterized AFE polygons. Despite the simplicity of the model it showed a fair to good model fit, with a 10-fold cross-validated area-under-the-receiver-operating-characteristic-curve (AUC) value of 0.89.

Then, the GLM model was used to predict the suitability of gray alder from 17 ka till today in 500-year steps with 5 km resolution and Lambert azimuthal equal-area projection. Glaciated areas were defined as unsuitable and were taken from the CHELSA-TraCE21k glacial reconstructions. We used the resulting time series of climatic suitability as input to the KISSMig (Keep it simple stupid migration) model (Nobis and Normand, 2014), which iteratively uses a simple $3 \times 3$ cell algorithm to calculate the spatial spread from a given origin from 17 ka to present. Presences and absences were weighted equally for the initial GLM calibration, and

KISSMig used squared suitability values to fulfill basic empirical expectations (see http://purl.oclc.org/wsl/kissmig, last access: 30 December 2019).

We tested for each AFE polygon of the current gray alder distribution all $25 \times 25$ km areas across Europe as potential refugia. All $5 \times 5$ km grid cells of those areas suitable at 17 ka were kept as refugia if the respective AFE polygon was accessible, and the spread pattern generated the lowest number of false positives when compared to the current AFE distribution. Because the migration ability of gray alder was unknown a priori, KISSMig simulations used 1 to 10 iterations for each 500-year step, corresponding to a maximum migration rate of 10 to $100 \, \text{m a}^{-1}$. For each iteration number, the combined spread pattern from all detected effective refugia was compared with the current distribution based on F1 scores. The optimized iteration number was identified by optimizing F1, which showed for gray alder a maximum migration rate of $50 \, \text{m a}^{-1}$. For a comparison with genetic clusters (Dering et al., 2016), the locations of that study were linked to the detected effective refugia with the shortest Euclidean distance for simplicity.

Current genetic clustering of populations indicates that the modeling of *A. incana* distributions at 17 ka shows that simulations based on CHELSA-TraCE21k successfully detected glacial refugia in the southern Alps, southern Norway, northern Norway, the Balkans, and eastern Romania (Fig. 7). The situation in eastern Europe is more complex, with most refugia located in Russia. However, since we only used the current distribution of *A. incana* in western Europe the results might be biased towards the east.

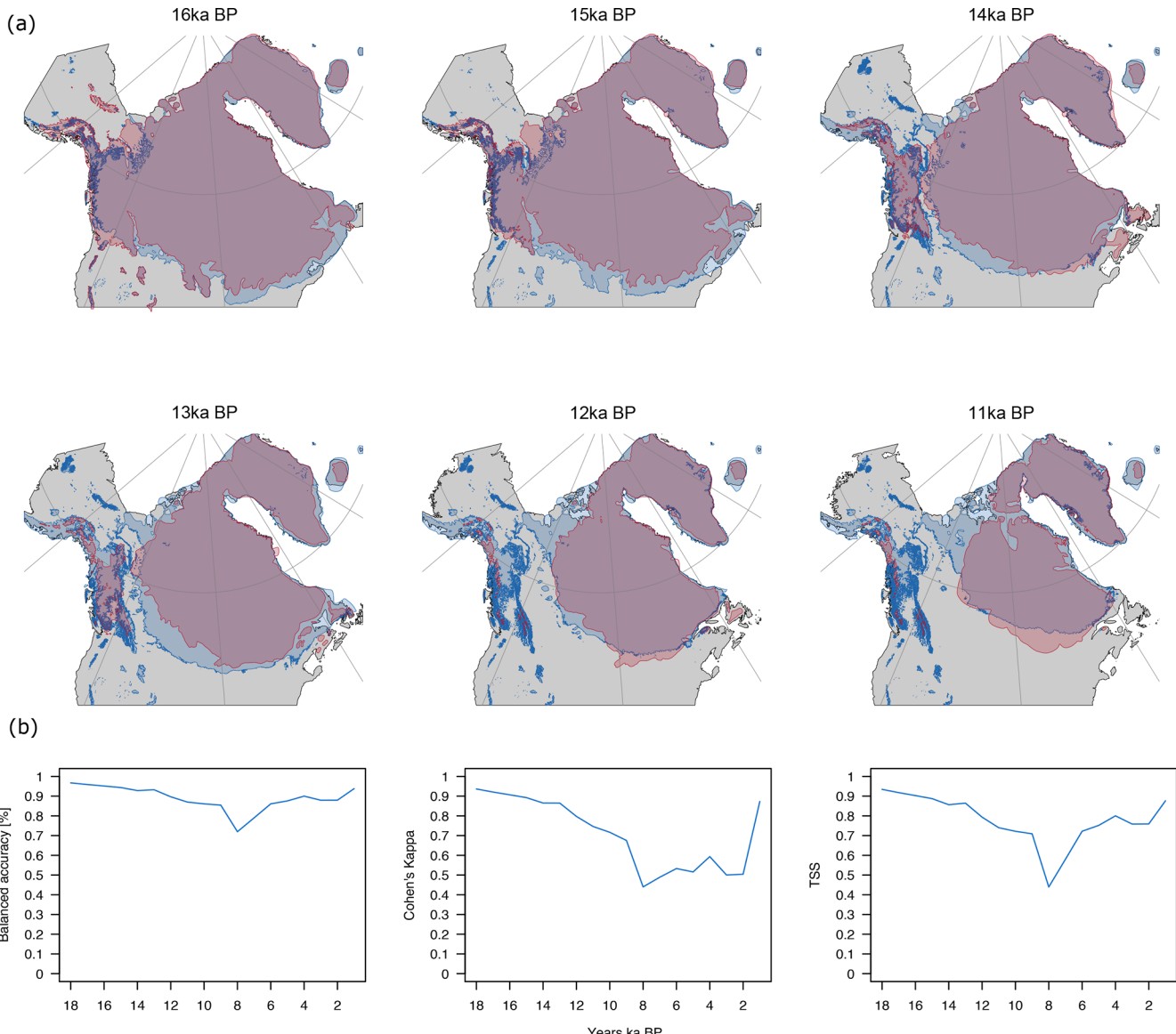

**Figure 6. (a)** Comparisons of estimated glacial extents of the Laurentide ice sheet from 16 to 11 ka. Blue delineates the interpolated ice sheet extent from CHELSA-TraCE21k, and red shows the estimated extent from Dyke (2004). While the retreat of the main Laurentide ice sheet is similar in both estimations, the Cordilleran ice sheet covering the Rocky Mountains retreats faster in the estimations by Dyke (2004) compared to CHELSA-TraCE21. **(b)** Performance comparison using three different metrics (balanced accuracy, Cohen's kappa, and true-skill statistic) from a comparison of CHELSA-TraCE21k and Dyke (2004).

## 6   Conclusions

Although both the original TraCE-21k and the downscaled CHELSA-Trace21k data track the relative temperature change well compared to ice cores, both models have relatively high temperature biases in absolute temperatures. Both the original data and the downscaled data have a warm bias before the Younger Dryas and a cold bias after it relative to the ice core proxy data. There are several reasons for this: coupled atmosphere–ocean general circulation models (GCMs) such as CCSM3 cannot provide regional-scale or unbiased information on a variety of climatic processes (Meehl et al., 2007). Temperatures from ice cores themselves are only based on proxy data, and the overall performance of such proxy data in estimating absolute temperatures is connected to biases themselves (Erb et al., 2018). The downscaling of the CHELSA-Trace21k data involves a trend-preserving (Hempel et al., 2013) change factor step to explicitly preserve the trends in TraCE-21k. If, however, these trends are already underestimated by the TraCE-21k data, they will also be present in the downscaled data.

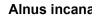

**Alnus incana**

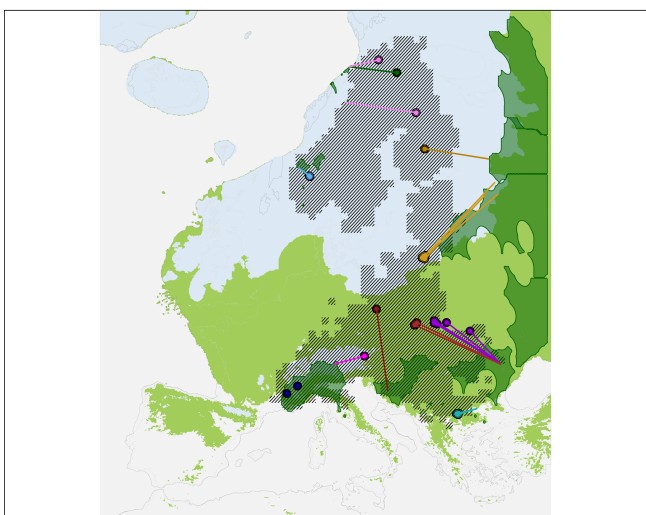

**Figure 7.** Distribution of *Alnus incana* in Europe (based on the Atlas Flora Europaea; Jalas and Suominen, 1976) in current times (line shaded/hatched) and reconstructed effective refugia at 17 ka (dark-green alpha hull polygons) using dynamic species distribution modeling based on KISSMig and CHELSA-TraCE21k. The entire suitable habitat for *A. incana* at 17 ka is indicated as light green. Although almost all of south-central Europe was suitable for *A. incana* at 17 ka, it might not have occurred at all locations due to dispersal constraints, which are considered in the dark-green KISSMig reconstructed distribution. Colored circles indicate the population genetic structure of *A. incana*, taken from Dering et al. (2016), where each color represents a genetic cluster. Lines indicate the most likely effective refugia a genetic cluster can be associated with, given dispersal and climatic constraints. Current genetic clustering of populations indicates that the modeling of *A. incana* distributions at 17 ka successfully detected glacial effective refugia in the southern Alps (dark blue), southern Norway (light blue), northern Norway (pink), the Balkans (dark red), eastern Romania (turquoise), and the Black Sea (dark red and violet). As we only use the current distribution of *A. incana* within the AFE extent the results might be biased outside of it.

The estimation of glacial extents shows an accuracy of $> 80\%$ compared to expert delineations of the glacial extent of the Laurentide ice sheet. There is, however, a clear drop in accuracy at the 8 ka event, when atmospheric methane concentration decreased, leading to a cooling and drying of the Northern Hemisphere (Kobashi et al., 2007). The strong coupling of the ice interpolations with only temperature might cause the decrease in performance as the downscaling algorithm ignores changes in precipitation that are only present in the driving ICE6G data. As the downscaling assumes an increase in glacial boundaries with cooling, this effect might not be realistic under an overall drying climate, and the fast shifts in temperatures over only 150 years (Kobashi et al., 2007) might also not be well represented in a model with 100-year resolution. Another problem in the esti-

mation of the glacial extent might involve errors from the applied *B*-spline interpolation. The resulting ice cover from this interpolation can, in some areas, only be a few meters thick, not representing real glaciers, but rather a spatial autocorrelation artifact of the interpolation approach used (e.g., see Fig. S1; 13 ka). Another source of error is that changes in bedrock due to the release of pressure from the melting ice sheets are not yet included in the algorithm. This can potentially result in several hundred meters of bias in affected areas that have not been taken into account in the current version of the algorithm.

The CHELSA-Trace21k data seem to be able to recreate the distribution of temperature and precipitation in a meaningful manner so that the use of the data in subsequent analysis produces meaningful results. The reconstruction of the refugia for *Alnus incana* shows that the combination of high-resolution climate data with a dynamic distribution model was able to accurately detect refugia, even those of a few kilometers in extent (Parducci et al., 2012), which cannot be detected using coarse climate data.

**Code availability.** Downscaling codes are based on Karger et al. (2017a), and all modules used are open source and integrated into SAGA-GIS, available here: https://sourceforge.net/projects/saga-gis/ (Conrad and Wichmann, 2015). The code unique to this study is written in R and creates the paleo-orography and glacier interpolations and is also available on Zenodo (https://doi.org/10.5281/zenodo.4545753, greenmind1980, 2021).

**Data availability.** All post-processed data and additional input files other than those provided by TraCE21k can be accessed at envidat.ch (https://doi.org/10.16904/envidat.211, Karger et al., 2021a). The data are published under a Creative Commons Attribution 2.0 Generic (CC BY 2.0) license.

**Supplement.** The supplement related to this article is available online at: https://doi.org/10.5194/cp-19-1-2023-supplement.

**Author contributions.** DNK, MN, and NZ developed the idea. DNK developed the model and implemented the code. DNK, MN, and SN validated the data. NZ, SN, and CHG funded the project. DNK wrote the first version of the manuscript, and all authors contributed to subsequent revisions.

**Competing interests.** The contact author has declared that none of the authors has any competing interests.

**Disclaimer.** Publisher's note: Copernicus Publications remains neutral with regard to jurisdictional claims in published maps and institutional affiliations.

**Financial support.** CEI This research has been supported by the Swiss Federal Institute for Forest, Snow and Landscape Research (ex-CHELSA and ClimEx grants); BiodivERsA CO-FUND (FeedBaCks grant); the Swiss National Science Foundation (grant nos. 20BD21_193907, 20BD21_184131); the ERA-NET BiodivERsA–Belmont Forum (FutureWeb grant); the Swiss Data Science Center (SPEEDMIND and COMECO grants); and the Aarhus University Research Foundation.

**Review statement.** This paper was edited by Irina Rogozhina and reviewed by two anonymous referees.

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
