# Peer review of "CHELSA-TraCE21k – High resolution (1 km) transient temperature and precipitation data since the last glacial maximum"

_Climate of the Past, 2021_

## Author Response (AR1)

**REVIEWER 1**:**

Review of "CHELSA-TraCE21k v1.0. Downscaled transient temperature and precipitation data since the last glacial maximum" by Karger and others

Summary
* * *
The manuscript describes an approach to produce high-resolution climate data by downscaling the output of a long simulation of a general circulation model using additional data sources. Most of the text is rather technical, a description of the downscaling process and validation of the resulting temperature and precipitation data. The paper ends with a potential use case of the produced dataset, the application to a problem in paleo-biology.

General comments
* * *
Overall, I appreciate the manuscript as an interesting contribution to facilitate paleo modelling work, which relies on high resolution past climate data. In my view, however, there is a number of severe shortcomings in the paper that need to be addressed before it can be published.

My first question was if the manuscript is well placed in the context of CP. The manuscript largely reads like a model description paper and may be better placed in a journal specialised for such content (e.g. GMD). The comments I will raise further on will not depend on this decision. But I will suggest revisions that bring out the modelling aspect even more, asking for further details that are currently lacking.

**Response: The manuscript has actually been transferred from GMD from the editors as the fit was considered higher with CP.**

The manuscript is giving a good overview of 'what' is done, but has severe shortcomings in explaining 'why' and 'how'. I believe there is need to improve on describing the motivation for most of the decisions and clarifying the details of the processes (see specific comments below). The use of symbols is confusing and inconsistent and should be improved. The aim should be to put interested readers in the position to understand and reproduce the work that has been done. Additional figures/illustrations may help to achieve that.

Response: We included two new figures highlighting the downscaling procedure in the revised manuscript. The figures are highlighting the procedure. The reproducibility is given by the publication of the source code of the model as well.

There may be a conceptual problem with the reconstruction of past surface elevations for glaciated regions. It is not clear to me why past sea surface elevation (i.e. global sea-level) is needed to correct the elevation (I109). The surface elevation of a glaciated region is the result of changing ice thickness and changing bedrock elevation. Neither of these changes is

related (linearly) to sea-level changes. A better explanation is needed to justify the presented approach.

Response: The model we present includes the ice thickness from ICE6G as indicated in line 108. The sea level is included as the high resolution DEM is based on a current bathymetric DEM. Not including the sea level changes would result in an orography that still has today's sea level, which is incorrect for past time steps. The model does not include a changing bedrock on the high resolution (1km) elevation however. We will add a comment on this in the revised manuscript.

Line 168: "Although this approach includes changes in the glacial surface and sea-level rise, it ignores changes in bedrock elevation due to upwelling after glacier melt."

The same applies to the coupling with temperature, which appears to modify the elevation estimate. Again, this is not well motivated and described.

Specific comments
* * *
Abstract

I12 Suggest to start a new sentence after (ICE6G) and lay out in simple terms what the temperature coupling entails. This has not become clear to me throughout the manuscript. The amount of ice at a certain place is not determined solely by local temperature, if that is what is happening here.

Response: The amount of ice is of course a balance of precipitation fluxes and temperature changes, which can be physically modeled with a numerical model. This is however not feasible at 1km resolution due to computational limitations and not due to the approach presented here. We use mean annual temperature as a proxy for an interpolation approach. The entire approach is still an interpolation and needs a local correction which we describe in lines 149ff. We see from the reviewer comments however that there have been several misunderstandings on the interpolation procedure and our focus on a mathematical description may not have been easy to understand. We therefore will also include a figure (Fig. 2) highlighting the interpolation algorithm that creates the orography better.

113 At this stage the reader will not know what CCSM3-TrCE21k and CHELSA stand for. This requires a bit more explanation already in the abstract.

Response: We will included a short description.

Line 13: TraCE-21k (Transient Climate Evolution of the last 21,000 years) based on the Community Climate System Model version 3 (CCSM3). Based on the reconstructed paleo orography, mean annual temperature and precipitation was downscaled using the CHELSA (Climatologies at high resolution for the Earth's land surface areas) V1.2 algorithm.

I16 Here the species distribution is described as a validation of the dataset, while later it is an application of the forcing data. Which one is it?

Response: Both. Paleoclimate models often get validated by specific proxy variables, as we are lacking direct measurements. Appling the dataset to model the distribution of species is the 'application'. Checking if the results make sense is not a 'validation' as such, but a 'plausibility test'.

I31 It is not clear to me how satellite data can be used to 'bride the gap between the coarse GCM output and the high resolution needed'. The two sources are distinct and have their own biases. Can you explain?

Response: Maybe this is misleading at this point. Deleted.

143 'ice shields' --> 'ice sheets'. Correct this also in the rest of the manuscript.

Response: Changed

143 'along the poles' --> 'in polar regions'

Response: Changed

I58 How does a simulation that starts at 21k-BP with 100 yr time steps come out at 1990?

Response: This is based on the timesteps of CCSM3 TraCE-21k. The last timesteps for the 20th century go until 1990.

IS9 Isn't paleo-orography an \*input\* to the downscaling procedure? Here it looks like an output.

Response: The output of one timestep (t) is the input for the next time step (t+1).

I62 It would be useful to distinguish between the model (CCSM3), the specific simulation (TraCE-21k) and the output of that model for a specific simulation (CCSM3 TraCE-21k).

**Response: Changed**

I69 Should add here that the model is run with a fixed topography (which one? PD, LGM) and fixed land-sea mask (if that is so). Is the fixed land-sea mask not a problem for the downscaling? How does your process deal with regions that change from land to ocean with deglacial sea-level rise?

Response: No, there is no fixed topography. The modeling of the paleo-ography is explained in paragraph 3.1 and 3.2. This is also why we include the sea level in the estimation of paleo-orography. The is now also a figure (Fig. 2) that describes this better.

173 What does the acronym CHELSA stand for?

Response: Climatologies at High Resolution for the Earth's Land Surface Areas. Included

I76 Explain what GPCC stands for.

Response: Included.

I78 ICE-7G appears to be available since 2018. Can you explain why you are using ICE6G? What is the difference between ICE6G and ICE6G\_C and why did you chose the 'C' variant?

Response: ICE6G\_C was the latest version when we made the calculations in 2017. ICE-7G was not available at that point. An update is not feasible at this point due to the high computational demand.

I78 Same point as above, is ICE6G\_C the model the simulation or the data?

Response: The output of the model (the data).

I80 To my knowledge ICE6G is not a dynamic ice sheet model and does not explicitly model changes in ice thickness.

Response: That is not what is stated on their webpage nor in the file metadata: "Each file contains information for points on a 1x1 degree global grid. The model's ice thickness field is given by the variable stgit." https://www.atmosp.physics.utoronto.ca/~peltier/data.php

I82 Topography update is every 500 years, but according to I57 climate updates every 100 years. How do you deal with this issue?

Response: This is indeed not described. The nearest time step has been used. We will add this at the respective point.

Line 225: As ICE6G has a 500 year resolution we used the ICE6G orography used that is closest to each timestep.

I85 It could be useful to explain here or elsewhere that the given extent mask is not necessarily in agreement with ICE6G at LGM. When is LGM defined in Ehlers et al., (2011). Is there a possible temporal mismatch with ICE6G?

Response: This is correct. We will mention this.

Ehlers et al. 2011 defines the LGM as 21 kBP (Page 10, Figure 1.5), similar to the definition used for the LGM in TraCE21k. ICE6G\_C has data that extends further into the past. We used the 21 kBP timestep in ICE6G\_C. We are defining it in more detail now:

"As the extent of the glaciers during the LGM (Last Glacial Maximum, hereafter defined as 21 kBP similar to Ehlers et al. 2011), we use data from Ehlers et al. (2011) that presents an upto-date, detailed overview of Quaternary glaciations all over the world, not only with regard to stratigraphy but also with regard to major glacial landforms and the extent of the respective ice sheets."

188 What year do you assign to this dataset, what does 'current' mean specifically?

Response: This cannot be clearly stated as GLIMS does not give this information unfortunalty.

I94 What does 'derived' mean. What is different and what is the same compared to the original CHELSA V1.2. Again, is CHELSA V1.2 the algorithm (as stated here) or the dataset originating from it?

Response: As stated.

The model is:

Karger, D. N., Conrad, O., Böhner, J., Kawohl, T., Kreft, H., Soria-Auza, R. W., Zimmermann, N. E., Linder, H. P., and Kessler, M.: Climatologies at high resolution for the earth's land surface areas, Scientific Data, 4, 170122, 2017a.

The dataset is:

Karger, D. N., Conrad, O., Böhner, J., Kawohl, T., Kreft, H., Soria-Auza, R. W., Zimmermann, N. E., Linder, H. P., and Kessler, M.: Data from: Climatologies at high resolution for the earth's land surface areas, https://doi.org/10.5061/dryad.kd1d4, 2017b.

I96 Why is GMTED2010 not described as input data in section 2? It should.

Response: Included.

"The Global Multi-resolution Terrain Elevation Data 2010 (GMTED2010) (Danielson and Gesch, 2011) dataset contains elevation data for the globe collected from various sources. Here we use the 30 arcsec. version of the data that represents the mean elevation of all 7.5 arcsec. grid cells that represent the highest available resolution of the data."

198 Why is the Miller data not described as an input dataset in section 2? It should.

Response: Included.

"We used data from Miller et al. 2005 for the estimation of global sea-level change from 21 kBP to 1990. The data provides global estimates of sea level change over the last 100 million years ago. The entire time series of sea-level change is based on a variety of proxy data, with the data used here, that dates back to the LGM, being mainly from dating using tropical reefs (Miller et al., 2005)."

1100 Could state here that the details 1-4 are described in the following sub-sections. It would be useful to describe the overall process in a flow diagram or other schematic to make it easier to understand the different steps. Add motivation at every step why things are done the way they are and how in detail.

Response: We will included two new figures (Fig.1, 2) in the revision and make sure to better describe each step in the respective paragraphs in the text. Each paragraph describing a step is also included in Fig. 1 so that it is easier for the reader to follow both figure 1 and the text. The algorithm is however, still complex, but we hope it is clearer now.

1102-104 It seems that this part still belongs to the general intro section 3 assuming that 3.1 is only about orography.

**Response: We rearranged this.**

1105 Explain what the purpose of combing Ehlers and ICE6G is. Motivate this by laying out your assumptions (do you trust Ehlers more than ICE6G in terms of accuracy?). What does Ehlers give you that ICE6G doesn't and vice versa? It may be useful to illustrate the whole process with a figure for one or several example location. Maybe a cross section through the margin of an ice sheet?

Response: We will included a new figure (Fig. 2). Ehlers gives a high resolution that can be used to delineate glacial boundaries at 1km. ICE6G gives a temporal signal of the elevation only of the major ice sheets but does not have a high resolution.

1106 Explain the choice and significance of taking 100 samples? How many samples are left (on average, max) after removing the outliers. Is ICE6G distributed at 1 degree resolution? If so mention it in section 2.2. If not, why work at that resolution?

Response: 1° the native resolution. 100 samples is arbitrary in this case. It seemed a good compromise between size of the resulting sample and accuracy.

I106 Not clear what "extracted the height of the glacier plus the surface elevation" means. Maybe 'height of the glacier' is ice thickness? Or is it height as in surface heigh? Is  $e_{t}^{ice}$  the surface elevation or the surface elevation + glacier height?

Response: We clarified this and also made sure that we use orography, topography, and glacier thickness more distinct.

1107 What does the subscript 't' stand for?

Response: Timestep. Clarified.

1107 Explain why these points are omitted. What is the reason for further extracting the point locations on the boundaries?

Response: Otherwise there would be a surface elevation of a glacier where there is no glacier.

1108 What does 'DEM' stand for and what is the data source for it?

Response: Digital Elevation Model. Changed. Data source is GMTED2010.

1109 Is 'past sea surface elevation' from Miller? Explain why this correction to sea-level is needed. As mentioned in the general comments, I don't understand why this is done and suspect a conceptional problem. Please explain this.

Response: Please see our response above. We don't understand what the conceptual problem could be, please elaborate what you expect. Our intent was to a) provide evidence for land where there was land in the past, and b) to use accurate and realistic orography data for downscaling the climate data from TraCE21k.

I116 What does subscript 'c' stand for.

**Response: cell. changed.**

I have tried to give exhaustive comments on page 4 to show the level of detail that is in my mind required to make this a useful description. Similar comments could be made in the sections that follow.

1120 Explain up front why the B-spline interpolation is needed and what the main ideas of the iterative approach are.

Response: A B-Spline interpolates data to higher resolutions, the iterative process (hence: "Multilevel" B-Spline) makes sure that the error of the interpolation decreases.

1131 What is this 'change factor'? Explain what it serves for in the approach.

Response: To remove the bias in the GCM as already stated in the sentence.

I131 What time period is tas\_{cur}^{mod} averaged over? Why do you resample to 0.5 degree resolution?

Response: Changed

1138 What is the significance of levels 26 and 20. What pressure/altitude do they represent?

Response: They are close to the surface and are needed for the CHELSA V1.2 alrgoithm . We tried to better explain this.

1146 The main ideas of that temperature coupling process have to be explained. What assumptions go into that approach? How is temperature assumed to modify orography?

Response: The assumption is that increasing temperatures are related to melting of glaciers (and vice-versa).

1155 Where does 'glacial melt' come from in this approach?

Response: Probably better explained as reduction in glacier extent.

1166 What process is assumed to modify orography?

Response: In this case only the changes in size/thickness of the glaciers. It is explained in step 3.1

1195 in 1184 the grid is described to have 4km resolution. Why the change to 3km?

Response: Simply a typo, Corrected.

l225 text here appears to be repeated in l227.

Response: Corrected.

Figure 1

Suggest to (additionally) show anomalies relative to the present day. For now, it is difficult to make out clear differences in these plots.

**Response: The figure has been removed**

Suggest to show the present day reference temperature field for comparison.

Response: The figure has been removed

The '-' in 22k-BP should re removed. It reads like a minus sign.

**Response: Changed throughout the text**

Is this a perceptually uniform colour map? If not, consider using one (e.g. https://www.nature.com/articles/s41467-020-19160-7)

Response: The figure has been removed

Results should be shown relative to a long-term average rather than one year (1990). If this is the case, what period is the data averaged over?

Response: 1950-1990 as this is the only period where we have a overlap between models and observations.

1288 Motivate why it is needed to project the data to another map? Details like projection parameters can be presented elsewhere (table, appendix).

**Reponse: Excluded**

I290-294 What is the underlying physical assumption for the 1/0 assignment? Clarify.

Response: 0 means there is no glacier, 1 means there is glacier. This binary information is then tested. Why does this need a physical assumption? Please elaborate.

1314 Is the strong correlation maybe related to the fact that the data was bias corrected to a similar product? How do you explain such impressive match?

Response: The bias correction certainly has an influence as well, but so does the downscaling. The CHELSA model decreases the bias and increases the correlation usually (for more details see. e.g. Karger et al. 2017, 2020, 2021 Scientific Data);

1334 'idiosyncratic' Strange choice of word. Reformulate?

**Response: Changed**

**Figure 5**

The strong mismatch at LGM could suggest that the lapse rate correction plays out in an unexpected way. It should be checked if that mismatch arises from climate model bias, lapse rate corrections or the bias corrections that are applied in the process.

Response: It could be all of the above plus inaccuracies in the proxies. It cannot be disentangled at this point, as we have no data to check the degree of bias in LGM climate simulations (the coarse resolution model output, which is input to the downscaling algorithm) or the proxy data.

We are now including this statement in the discussion.

"The bias observed after downscaling might be related to factors to a bias in all the different input sources, such as the TraCE21k bias being amplified, a bias in the ice-core proxy data itself, or the bias correction using the relatively simple change factor method. With the available data, these potential causes cannot clearly be disentangled, but should be kept in mind for applications of the data."

I344 It is not clear to me how glacier extent is meant to validate the downscaling process. It may serve to validate ICE6G and may reveal a mismatch between ICE6G and Dyke, but that is not really at stake here. Could you explain how that comparison can constrain your approach? How does unmodified ICE6G compare to Dyke. Is that improved with your modifications?

Response: We compare the 1km predictions estimated using our algorithm, not ICE6G at 1° resolution. The 1km predictions of glacial extent are the most derived parameter from our dataset with the highest uncertainty attached to it (as it requires several input parameters with high uncertainty itself).

I360 This section comes with unexpected new concepts and models (GLM, KISSMig) that were not introduced before. After going through the technicalities of the sections before, this is a steep change of register. In the abstract this part is introduced as another aspect of the model validation, while here it is written as a use case for the produced dataset. In either case, I suggest this part has to be better linked with the rest of the paper or, maybe better, extended and conceived as a separate paper.

Response: Its part of the validation with the idea that it simultaneously presents a case study as a plausibility test. We will add more basic information about the reasons for applying this case study and about the underlying models used such as KISSMig.

I381 Why is yet another projection needed in this case? Motivate.

Response: Since KISSMig is a grid based dynamic species migration model, it uses an equal area projection to avoid spatial bias in the simulations.

I418 What trends are to be preserved? Clarify. Reference to (Hempel et al., 2013) is probably better placed in the description in section 3.2.

**REVIEWER 2:**

CHELSA-TraCE21k v1.0. Downscaled transient temperature and precipitation data since the last glacial maximum", Karger et al.

**General comments:**

This manuscript presents a method of downscaling climate model data, using an algorithm, with objective to create high-resolution global monthly climatology for precipitation and temperature for the period of last 21000 years. The manuscript consists of description of the method, evaluation of obtained results and an example of potential use of created high-resolution climate data set in paleo-ecology to detect refugia of plant species at the end.

The final product of this research, in format of high-resolution climate data set, presents an important result, that should hopefully find its application in several scientific fields. I highly appreciate efforts to get the final data set, which, I believe, was a difficult, time-consuming and very technical task. That technical part is well presented and documented within the manuscript. However, I find that the rest of the manuscript has serious shortcomings, especially the evaluation of the obtained data set, what is expected to be the central part of this research. There are also serious issues in manuscript structure. Some figures are presented without any deeper analysis, while on the other hand there are chapters that describe validation of the obtained data set, but without figures, that actually follow in subsequent chapters, which affects significantly the readability of the manuscript. At the and, there is a well described chapter about potential application of the high-resolution data set. however, without clear connection with the rest of the manuscript. It could represent a highlight of this research and be a crucial proof for efficient application of the data set, but it is not even mentioned in the main objective (although it exists in abstract and is mentioned in introduction). There are also inconsistencies in use of terms and symbols throughout the text that, for example, lead to confusion in understanding of some parts of the manuscript, of some figures and even in understanding of correct name of the algorithm in the abstract.

This manuscript has some good material and important results, but it requires significant improvements and better structure in order to be considered and at the end accepted for publication in this journal. Therefore, I would suggest a major revision, to give the authors a chance to improve it, but with caution to stay within the scope of this journal. My further specific comments are listed as follows:

Response: Thank you for your judgment on the value of the generated climate datasets. The generation of this dataset was our main aim, and the evaluation of its usefulness will be improved.

Specific comments:

Line 9, Line 14: What is the name of the algorithm? Is it "CHELSA-TraCE21k downscaling algorithm" or "CHELSA V1.2 algorithm"? Please, be consistent in using specific terms throughout the text. In Line 17 it says "CHELSA TraCE21k output" (without hyphen), which leads to confusion since the very beginning. In addition, in the title of the manuscript it says: "CHELSA-TraCE21k v1.0", and that part "v1.0" does not appear at all in any part of the manuscript.

Response: Will made several modifications and tried to be more consistent by better explaining the terms and by cleaning the text accordingly.

Lines 26-29: There are several applications mentioned, where temporal and spatial variability of temperature and precipitation matter. I would like to see at least one more of these applications described in detail, where your high-resolution data set can be used. I believe it could demonstrate the added value of created high-resolution data set. However, that would probably lead to writing of a completely new manuscript, possibly out of the scope of this journal.

Response: It seems odd that this is mentioned here, as we gave one application an entire chapter (Chapter 6). We are not sure how we can add more here. We added a short paragraph on other applications that are already accepted or published and use this dataset (as it is available already for quite some time). Performing such applications would render the manuscript too heavy and hard to read. It would also deviate readers from the main aim of the manuscript.

Line 58: How do you end up with the year 1990, when you start from 21K BP and use 100years time steps? And, please, do not use hyphen in "21K BP", it is not correct, unless it is an adjective.

Response: This is based on the timesteps of CCSM3 TraCE-21k. The last timestep for the 20th century goes until 1990.

Lines 60-72: What is the main reason to use exactly this model? Please, justify.

Response: The model is transient and can therefore also be used for dynamic modeling, as for example highlighted in chapter 6. Additionally the data is readily available. It needs to be noted that we did not run the CCSM3 simulation ourselves, but simply downscale the model output.

Lines 67-72: If you say in Lines 66-67 that CCSM3 is global climate with coupled ocean, atmosphere, sea-ice and land surface components, then try to maintain the same order of the Earth system components when you describe characteristics of each one, in order to maintain consistency.

**Response: Changed.**

Line 74: What is CHELSA? Is it a data set or an algorithm? It is very confusing. What does this acronym stand for?

Response: Climatologies at high resolution for the earth's land surface areas. It's both the name of the output dataset and the algorithm. Explained in the revised version.

Lines 245-248: Very confusing, at the and, I don't understand what is presented in Figure 1. Especially due to use of hyphen in the figures, that gives impression it is a "minus" (22k-BP). Please, avoid that in all other figures, too. Also, there is no any discussion about that figure, only the statement in the legend that it shows "exceptional climate dynamics". Please, avoid use of such strong words, especially if they are not supported by any explanation.

**Response: We removed the figure. Instead we now present a figure showing the steps of the algorithm.**

Lines 258-264: It is not very clear what is shown in this figure. Did you calculate difference of all mentioned 100-year BP periods from 1990 year only? Or from some annual mean of 1960-1990 period, or 1900-1990? Also, there is no any discussion about this figure and the same comments stand as for the previous one. In addition, I see some strange separation in anomaly sign in southern hemisphere, approximately around 10 S and 40 S. Is there maybe some problem with the downscaling algorithm for that region? Or there is some physical explanation for this pattern?

**Response: The anomalies are actually taken directly from the CCSM3-TraCE21k dataset and therefore cannot be caused by the downscaling approach itself.**

Lines 265-359: I would suggest to reorder and rewrite chapters 4.1, 4.2, 4.3, 5.1, 5.2 and 5.3. In a current form, it is difficult to follow. It would look much better and improve readability if you could merge 4.1 with 5.1, 4.2 with 5.2 and 4.3 with 5.3.

Response: Thank you for the suggestion. We will reorder the manuscript accordingly by merging the respective chapters.

Line 266: What is the resolution of GHCN? Is it comparable with your data set?

**Response: GHCN consists of point measurements.**

Lines 360-409: This whole chapter does not seem to have a good connection with the rest of the manuscript, although it gives an important application of the obtained high-resolution data set. A suggestion could be to remove it from this manuscript and to try to improve the rest with more profound and more comprehensive evaluation of the data set. Current chapter could be used with several other examples of potential applications of the high-resolution data set with objective to create another manuscript.

Response: In the beginning of the review you stated that a potential highlight for an application would be needed. The chapter is included to show exactly such a use case. The application also serves as a plausibility check of the downscaled data. If the glacial refugia cannot be modeled correctly, the downscaled data might be wrong, which is not the case here.

Technical corrections:

Line 25: Spatial resolution should not be expressed in square kilometers. I would rather say "at spatial resolutions lower than 1 km", for example

Lines 73 and 78: Repeated chapter number

Response: Changed.

Line 76: Acronyms ERA and GPCC are mentioned for the first time in manuscript, therefore, it is expected to write their meaning.

Response: Changed.

Line 177: It seems there is an extra space between the words "level resulting"

Response: Changed.

Lines 184, 186, 195, 196: Please, use the hyphen when you have number, followed by unit when it is an adjective (4-km grid resolution, 3-km grid cell) and spacing when you have number and unit when it is not an adjective (1 km, not 1km). Try to maintain consistency throughout manuscript in all other similar cases.

Response: Changed.

Line 210: It is not understandable, is it a continuation of the sentence, that should be separated by comma and followed by small letter or something else?

Response: Will be changed.

Line 215: One "being" extra, please, remove it.

Response: Will be changed.

Line 216 and 240: 30-arc sec. resolution/grid

Response: Changed.

Line 229: windward-leeward equations

Response: Changed.

Line 234: Are these 2 dots instead of comma? Please, correct it.

Response: Changed.

Line 245: 1-km paleoclimatic dataset

Response: Changed.

Lines 255 and 263: Why is it written "8.2 kiloyear", when in all other cases you use only "k"? Please, maintain consistency in use of symbols throughout the manuscript.

Lines 282, 344, 373, 380, 395, etc.: 18k PB and 1k PB. Please, maintain consistency throughout the manuscript by correcting other similar cases

Response: Changed.

Lines 302-303: Another different way of writing 18k BP and 1k BP; 1-km resolution

Response: Changed.

Line 305, 308 and 316: It is RMSE, not RSME.

Response: Changed.

Line 320: Taylor diagrams; typing error with an extra "f"

Response: Changed.

Lines 328-343: "CHELSA\_TraCE21k model", "CHELSA\_TraCE", "TraCE21k", "TraCE", "CHELSA-TraCE21k time series data", "CHELSA V2.1" - so many similar and confusing names in this short paragraph, that it is impossible to follow. Please, rewrite it and try to be consistent in using specific terms.

Response: Changed.

Lines 354-359: ice sheet, not ice shield

Response: Changed.

Line 390: Typing error "the"

Response: Changed.

Line 420: "comparably well when compared". Please, try to find better words

---

## Referee Report (RR1)

The latest version of the manuscript leaves much better impression when compared with the previous one. I appreciate authors' efforts to improve the manuscript based on both reviewers' suggestions. It obviously required a lot of work, but the manuscript now has significantly improved structure and consequently, notably improved readability. Also, a newly added validation chapter crucially contributes to the quality of the manuscript. However, I still find there is a space and need for further improvements, so my observations and suggestions in this regard are as follows:

1) The title does not correspond adequately to what is presented in the manuscript. It seems to me that the data set is developed with intention to be used in paleo-ecology (at least the introduction indicates so), therefore, there is no reason not to put that into the title. Also, I would remove "V1.0" from the title, I find it useless. I would rather put "high-resolution data set" or even more specifically: "1-km data set" if that is something what will differentiate it from other potential similar datasets. A suggested title could be: "CHELSA-TraCE21k high-resolution (or "1-km") data set - downscaled transient temperature and precipitation data since the last glacial maximum – development, validation and application in paleo-ecology", or something similar, maybe shorter. If you want to maintain the current title, then I am afraid you would have to change the bigger part of the introduction.

2) It was really an unfortunate decision to name equally both the data set and the algorithm/model - CHELSA V1.2. Regrettably, I don't see significant progress in clarification in that context throughout the manuscript. Only between the lines 60 and 121, a reader can find the next phrases: "CHELSA V1.2 **algorithm**", "CHELSA V1.2 **climate data set**", "CHELSA V1.2 **mechanistic downscaling model**", "CHELSA V1.2 **procedure**", "CHELSA **downscaling model**", "CHELSA V1.2 **model**". It is just unacceptable and extremely confusing. I find it necessary to add a paragraph, for example, somewhere at the beginning of chapter 2, clarifying that and informing the reader there are the dataset and the algorithm/model with the same name. It has to be very clear. And try to use only 2 words in its description throughout the text, for example "data set" and "model/algorithm". In addition, once again, please, maintain consistency throughout the text and decide whether you want to use "TraCE21k" or "TraCE-21k".

3) In abstract, you say:"*High resolution, downscaled climate model data are used in a wide variety of applications across environmental sciences*". Then in chapter 6, you start with:"*Transient long-term climatic data have a wide range of possible applications*". Please, add 2-3 examples where exactly, for example, just continuing the sentence by:"*..., such as A, B and C, for example*".

4) Are there other comparable downscaled data sets available on the market? If yes, then, what is the advantage of CHELSA-Trace21k, why is it different/better then the other ones, why it should deserve attention, why is it unique? Please, clarify in a sentence/paragraph, in conclusion, for example.

5) Before submitting a final version of the manuscript, I suggest a thorough inspection regarding the consistency of the use of terms throughout the manuscript, as well as typing and other errors

**Specific comments**:

Line 8:

High-resolution

Lines 9-10:

Here we introduce a new, high-resolution **dataset**, named CHELSA-TraCE21k. It is obtained by downscaling TraCE-21k data, using CHELSA V1.2 **algorithm** with objective to create global monthly climatologies for temperature and precipitation at 30-arc sec spatial resolution in 100-year time steps for the last 21,000 years.

Line 18:

Validations show that CHELSA-TraCE21k V1.0 **dataset** reasonably represents the distribution...

Line 27:

GCM states for **general** circulation model, not global

coarser grain=>coarser resolutions

Line 34:

has been bridged, or had to be bridged

Line 59:

Here we present paleo-climatic data, downscaled from the **CCSM3_TraCE21k model output (or TraCE21k dataset)** to a 30-arc sec. resolution using the CHELSA V1.2 algorithm

Line 67:

in various parts of the world **from**

Line 68:

The TraCE-21k simulation has T31_gx3v5 resolution (…..)

Line 70-71:

Which resolution??

Please, specify the resolution per model component, for example CAM has resolution 3,75°x3,75°, which is important information for this manuscript, because you are downscaling atmospheric variables

Line 76:

It includes **mean monthly daily 2m mean**, minimum, and maximum temperature ?!

Line 78:

ERA stands for 'ECMWF Re-Analysis'

Lines 332-335:

Figure 2, axis labels not very clear, should be improved

Line 361, 364, 373:

RMSE, not RSME

Lines 444-450:

Figure 6, a) and b) missing on the figure; a) is also missing in the legend

Line 455:

...**if** the transient…

Line 498:

Separate Figure 7 legend from the text below

---

## Author Response (AR2)

**EDITOR**

Dear authors,

as you have seen, reports of the two referees have deviated in their attitudes and decisions with regards to your article. In response to the criticism in review 1, you have expressed doubts that reviewer 1 has read your manuscript carefully enough and that their criticism is well grounded. I have therefore asked you to prepare a detailed rebuttal letter explaining your grounds for such statements and a revised manuscript highlighting all changes that you have made in response to reviews 1 and 2. Once you have submitted both the letter and the manuscript, I will evaluate your arguments together with an independent reviewer and will make a final decision about the publication or otherwise.

Good luck and let us hope for a positive outcome.

Kind regards, Irina

Dear Dr. Rogozhina,

Thank you for your work on this manuscript. We have now prepared a point to point response to all concerns and comments by the two reviewers.

**Reviewer 2:** Seemed ok with manuscript and suggested mainly changes of the text and a more consistent terminology throughout the manuscript. We incorporated all of the comments this reviewer had.

**Reviewer 1:** Is mainly concerned about how we created the orography that we used for the downscaling algorithm. The concern is that we do not include changes in bedrock elevation (which we cannot at this point). Additional the reviewer asks for a validation of the glaciers, which is however, already included as section 4.1. but ignored by the reviewer. We include a detailed response to the critique below. Were the reviewer had a valid point, we incorporated the changes.

We hope that you can follow our reasoning in this respect and deem the manuscript suitable for publication.

Best regards,

On behalf of all co-authors

**Dirk Karger**

**REVIEWER 1**

**General comments**
* * *
I have reviewed an earlier version of this manuscript as REVIEWER 1. While I acknowledge important improvements in the current version of the manuscript, it requires major revisions to make it suitable for publishing. A few of my earlier comments have not been addressed, so I am reiterating them once more.

With the given documentation, it is still not possible to judge if the applied methodology is flawed.

This concerns 1) the way only sea level (and not bedrock changes) are used to calculate elevation changes and 2) the interpolation procedure using temperature as a proxy for ice sheet retreat. I have suggested possibilities to validate the methods with available reconstructions, which I consider fundamental to support the choices presented in this paper.

Response: Thank you for your comments. It is unfortunate that our previous explanation has not been clear enough regarding the interpolation procedure of the glaciers despite the new and additional figures to explain it step by step. We already replied that the model does not include bedrock changes but only sea level changes in our previous response. There is very little we can do about this at this point. It might be included in a newer version of the algorithm if we choose to rerun and improve the model or if it gets picked up by the community. We are aware that this model has limitations and cannot be perfect in every regard.

Unfortunately we cannot find your concerns about using temperature as an additional variable in the glacier ice sheet retreat interpolations in your specific comments, so we cannot comment on that matter. Additionally, it seems like the validation of the glacier section 4.3 seems to be largely ignored unfortunately too. As

Please find a detailed response to the specific concerns raised below.

Specific comments
* * *
112.

Repeating my earlier comment: ICE6G is not a dynamic ice sheet model. I have looked up the description for you: "the ICE-6G\_C (VM5a) model that is under discussion in this paper is based upon the 'GIA only' methodology. In this methodology the ice thickness history as a function of position is simply adjusted iteratively in order to satisfy all of the available constraints". Please reformulate.

Response: Changed to: "...and interpolations using ice sheet data (ICE6G)..."

126.

Here you state "climatic conditions at spatial resolutions < 1 km", but later (e.g. 48) it is "~1km". Should be made consistent.

Response: Changed

l27."run at much coarser grains" --> "run at much coarser resolution"

Response: Changed

I34."has be bridged" --> "has been bridged"

Response: Changed

l44. "on earth" --> "on Earth"

**Response: Changed**

148.

It is somewhat trivial to state that 0.043 SYPD is 25-fold lower than 1 SYPD. What would be an example of a typical 1 SYPD simulation?

Response: It is unclear what is asked here for. SYPD is a measurement of computational efficiency.

160.

Repeating comments for "time steps of 100 years from 21k-BP to 1990"

Here and elsewhere, this should be written as "21 kyr BP"

BP typically refers to years before 1950. If you go in steps of 100 from 21 kyr BP forward, you never end up at 1990. Either you used a different time step than 100 years, or a different starting point. That is why you have to modify this statement.

Response: Changed to ka BP

160.

(TraCE-21k) should be defined the first time you use it (outside of the abstract).

Response: Changed

l64.

It would be good to add some information about what topography and ice sheet boundary condition (topography, mask, albedo) was used to produce this simulation. Is the land-sea mask constant? Is the topography changing? If yes, is the topography change consistent with the ICE6G\_C reconstruction?

Response: This is in detail given in:

He, F.: Simulating Transient Climate Evolution of the Last Deglaciation with CCSM3, PhD - Thesis, University of Wisconsin Madison, Madison, WC, USA, 171 pp., 2011.

Liu, Z., Otto-Bliesner, B. L., He, F., Brady, E. C., Tomas, R., Clark, P. U., Carlson, A. E., Lynch-Stieglitz, J., Curry, W., Brook, E., Erickson, D., Jacob, R., Kutzbach, J., and Cheng, J.: Transient Simulation of Last Deglaciation with a New Mechanism for Bølling-Allerød Warming, Science, 325, 310–314, https://doi.org/10.1126/science.1171041, 2009.

The model setup is nothing we did here. We did not run the TraCE-21k simulations, so its not appropriate to repeat the methods here, which would be out of the scope of the paper.

169.

Is "DGVM" the name of the specific model or the shorthand for all dynamic global vegetation models (as line 71-72 may suggest)? Clarify!

Response: as stated: "...dynamic global vegetation model (DGVM)..."

170.

Is the "land [...] model" the same as the vegetation model? If not, introduce the land model here.

Response: It's the land component of CCSM3. We tried to clarify:

"...The TraCE-21k simulation was calculated at a T31\_gx3v5 resolution (Otto-Bliesner et al., 2006) using a coarse resolution dynamic global vegetation model (DGVM). The coupled atmosphere-ocean model in CCSM3 is based on the Community Atmospheric Model 3 (CAM3), on 26 vertical hybrid coordinate levels. The land and atmosphere components in CCSM3 in the TraCEF-21k simulations uses the same resolution. The parameterizations of the DGVM are largely based on the Lund-Potsdam-Jena (LPJ)-DGVM. The ocean model in CCSM3 uses the NCAR (National Center for Atmospheric Research) version of the Parallel Ocean Program (POP) with 25 vertical levels and the sea ice model is the NCAR Community Sea Ice Model (CSIM)..."

176.

What is "mean monthly daily 2m mean"? Maybe "monthly mean, minimum and maximum temperature and precipitation fields"?

**Response: Changed**

178.

You call it "mechanistic climate downscaling". I am familiar with terms statistical and dynamical downscaling. Which category does yours belong to?

Response: None of the above. It is a hybrid model, that has mechanistic components, but also statistical ones. Another term we commonly use is topographic downscaling.

Changed to: topographic downscaling

184.

Remove "and dynamics". ICE6G is not a dynamic ice sheet model.

Response: Changed

185.

"from the Last Glacial Maximum" What is the first available year in the time series?

Response: 26kyr BP? Why is this important here? We start our downscaling at the LGM as stated.

187.

LGM is already defined. Remove "(LGM)"

Response: We removed 'Last glacial maximum' and keep the abbreviation

188.

LGM is already defined. Modify description.

Response: We removed 'Last glacial maximum' and keep the abbreviation

189.

How "up-to-date" is this dataset from more than 10 years ago? Suggest to reformulate.

Response: deleted 'up to date'

195.

What time does "the 'current' extent of the glaciers" refer to? I understand that this may not be clearly defined by the dataset providers (late 90s, early 2000 maybe), but you are assigning it eventually to a certain time in your modelling (maybe 1990?). This should be mentioned here or elsewhere as in "We are assigning the dataset to the year xxx in our modelling". The same applies to all the other data sources (GMTED, GEBCO), which may be best done by a summary statement in the end.

Response: Unfortunately the GLIMS database does not give a reference year. We therefore use as referring to the reference period (1950-1990).

L105.

"we keep as land altimetric data that of the CHELSA V1.2 procedure" I didn't find a description what topography is used in CHELSA. Should be added in 2.2.

Response: Page 3, Line 13 in Karger et al. 2017, Sci. Dat. "...the Global Multi-resolution Terrain Elevation Data 2010 (GMTED2010)...". or Figure 1.

**We included it now:**

Although GEBCO also includes land surface altitude, we only use it for the oceans, and we keep as land altimetric data that of the CHELSA V1.2 algorithm (that being GMTED2010) to maintain comparable topography at the land surface.

**Section 2 in general**

Since this has become a long list of rather short subsections, it could be an idea to display the information about the different datasets in a table (name, description, time coverage, reference, ...) complemented with a summary paragraph.

**Response: This is a matter of personal preferences and we actually prefer to keep it as it is. I117.**

"As the orography at different time steps between 21k BP and current times is not available" I think that ICE6G\_C would give you that information. Maybe add "... at the high resolution required for our downscaling method" to make sense of this statement.

Response: Changed to: "...As the orography at different time steps between 21ka BP and current times is not available at the high resolution required for the CHELSA algorithm..."

**Figure 1, 1124.**

Terminology. I am not an expert on this, but it seems that the distinction made here between topography (as relative to present day sea-level) and orography (relative to current global sea-level) is not supported by common definitions of the two terms. Also, I could not find a description of bathymetry in the ocean and surface elevation over land in one generalised term. I would suggest to using your own symbols and describing what they mean, rather than using established terms that mean something else.

Response: There seems to be a misunderstanding: Orography generally refers to terrain above water (including glacier surfaces), topography can also include terrain under water, bathymetry contains terrain under water. This is the same terminology as used in ICE6G\_C (e.g. orog and topo)

**for example.**

I doubt that the combination of GMTED2010 and GEBCO really gives you bedrock topography (upper right in figure 1). I think GMTED2010 provides surface elevation over glaciated areas (Greenland/Antarctica), which is the upper ice surface. The bedrock topography is a few thousand meters below that. The same applies to 'bedrock' in the downstream box 'bedrock orography' to the left.

Response: That is correct, it gives the bedrock for non-glaciated areas. For the approach used here, this is however does not constitute a problem, since we are interpolating between past and current glacier extent. We clarified this by adding:

"...To create a bedrock orography  $e_t^{bed}$  (i.e. topography adjusted for sea level without glaciers except for currently glaciated areas)..."

The idea to correct present-day topography/bathymetry with global sea-level does not make sense to me. Surface elevation over this time scale does not only change due to changing sea-level, also due to isostatic changes of the bedrock. In the periphery of the ice sheets where it may be most important for your biological application, the bedrock change may well be the dominant signal. I see in 1169 that you acknowledge that bedrock changes are not taken into account. But why not?. ICE6G\_C will give you a consistent set of data for sea level and bedrock elevation, in addition to ice thickness. In fact, the bedrock change is likely the most reliable output of ICE6G\_C, because ice thickness is prescribed to get the right loading history. Why are you not using it? If you think your sea-level correction method gives a better representation of surface elevation, you should at least compare your results (with appropriate figures) against ICE6G\_C. This will show if your method is an appropriate approximation for the full solution. My intuition is that surface elevation will be off by a few hundred meters in proximity of an ice sheet. If I am overlooking something obvious here, please explain better in the manuscript why you chose to not use the full information provided by ICE6G\_C.

Response: We already replied to his point and pointed out that this is based on a misunderstanding from the reviewers side. We do not change the bedrock topography, and that it is not possible with the current algorithm applied. We simply make sure that the orography (terrain elevation above sea level) we use for the downscaling algorithm is actually adjusted to the sea level. If we would not do that, the land surface would not have changed in the last 21 thousand years due to increasing sea levels.

The reviewer mentions: "The idea to correct present-day topography/bathymetry with global sealevel does not make sense to me. Surface elevation over this time scale does not only change due to changing sea-level, also due to isostatic changes of the bedrock."

Why the correction of topography with sea levels does not make sense to the reviewer eludes us, since we did have major changes in sea levels over the last 21k years. That the model applied does not contain bedrock changes we already explained, and there is unfortunately nothing we can do about it at this point, except for mentioning that this effect is not included. We do not see how this is a problem. As an example: Most global climate models do not resolve convective precipitation. But that does not mean that these model are 'flawed', just because it does not include all possible processes.

**1128.**

The following description still misses clear motivations for why things are done the way they are done. You should start by making clear what information you have (e.g. present day seabed and land

topography), what you are trying derive, and how you are making approximations/interpolations to get there.

Response: We are not sure what you are missing here. The whole purpose of this section is to describe how we derived the paleo-orography. There are two additional figures even (Fig. 1, Fig. 2) that highlight the process. The equations are all given ('the way it is done'). What exactly is missing here?

**l130.**

I think from "We first combined ..." you are no longer at the LGM. That would be good to make clear. E.g. with "that provides the surface elevation e at the present day"

Response: Changed: "...We first combined the topographic information from GMTED2010 on land, and that of GEBCO into a bedrock topography that provides the current bedrock topography  $e_c^{topo}$  (including current day glaciers, see ff.)..."

**l136.**

Back to LGM? Confusing!

I think from l136-l147 (maybe even l149) you are describing how to produce a smooth ice surface elevation for the LGM on a high-resolution grid. Could be good to say that upfront.

The then following transition into a time-dependent estimate is confusing to me, because we don't know at this point how the interpolation between LGM and present will work. Maybe it would be better to leave the time dependence out of this for now?

Response: We describing first how we create the initial high resolution ice surface at the LGM (t=0) and then how we use this as a basis for the next timestep. (t0 ... tn). We don not see why it would be necessary (or even correct) to remove the time dependence here.

l170.

Without any validation/comparison against reconstructed glacial extents from the literature, it is not possible to judge whether this interpolation approach is an innovative method or a bad idea. See also comment l220.

Response: Why is section 4.3 ignored here? "Validation of glacier extent between 18ka BP and 1ka BP" ? Has this entire section been ignored?

|171.

"As high-resolution estimates of glacial surface elevation are not available for timesteps t other than the LGM"

For the LGM you had Ehlers2011 to delineate glacier extents, but the resolution of the surface elevation data is the same for all time steps in ICE6G\_C, isn't it?.

**Response: Yes, it is all the same: 1°. Why is this a problem?**

I173."at each time step t ≠ 0"But also not at present day, right?

Response: Correct. Changed to:

"...at each time step  $t \neq 0$  and t = 221..."

1181.

**"resampled to a 0.5° grid resolution"**

Can you motivate the need for this intermediate grid? Why not interpolate one of the products to the grid of the other?

Response: This is the same resolution as in the CHELSA V1.2 algorithm. We kept it constant with the algorithm.

We added: "...The resolution of 0.5° follows the same procedure as used in CHELSA 1.2 (Karger et al. 2017)..."

**1220.**

There are large data collections available to constrain the LGM-present glacial extent based on geomorphological constraints (e.g. https://doi.org/10.1016/j.quascirev.2015.09.016; https://doi.org/10.1016/S1571-0866(04)80209-4). How does your approximation compare to those reconstructions? Please include a validation for your method.

Response: We already included an validation of the glacier reconstructions. Why is yet another one needed? We already show deglaciation on the north American ice sheet in Figure 6. that is the basis of our validation. It is not clear to us why this section consistently ignored in this review?

**1240.**

'idiosyncratic'

I have commented this before and supposedly you had changed it. Here it is again.

Changed to: "...The CHELSA V1.2 algorithm assumes that orography is one of the main drivers of precipitation..."

1299.

Could you motivate the parameter choices for c and h? Have you tried other values? Are they taken from past experience with the model or from published values?

Response: They come from the tuning of CHELSA V1.2. See Karger et al. 2017, Sci. Dat.

1304.

Remind us what reference period the bias correction is calculated over and that you are using an intermediate grid.

Response: 1980-1990. Added

1331.

"tas being interchangeable for tasmax and tasmin in Eq. 22 and Eq. 23" This doesn't work, because you have already defined tas = (tasmax+tasmin)/2 in line 323. Need to find another symbol.

Response: Clarified to:

"...The downscaling of monthly near surface air temperatures (*tas, tasmax, tasmin*) follows the methods described in 3.2.2., with the only difference that instead of mean annual temperature, *tasmax* and *tasmin* are used, where *tas* = (*tasmax+tasmin*)/2. The temperatures have again first been bias corrected using:

$$\Delta tasmax_{m} = tasmax_{cur_{m}}^{obs} - tasmax_{cur_{m}}^{mod}$$

$$\Delta tasmin_{m} = tasmin_{cur_{m}}^{obs} - tasmin_{cur_{m}}^{mod}$$

$$and:$$

$$tasmax_{m_{t}}^{cor} = tasmax_{cur_{m}}^{obs} - \Delta tasmax_{m}$$

$$tasmin_{m_{t}}^{cor} = tasmin_{cur_{m}}^{obs} - \Delta tasmin_{m}$$

$$(22)$$

with *m* being the respective month of the year, in Eq. 22 - Eq. 25...."

(25)

**1335.**

Figure 2. It seems strange to switch region in the middle of the description from e) to f). It would be useful to continue with the first region to the end and add another figure for the second region if needed. For the validation of interpolated ice extent, it would also be good practice to show your reconstructed ice mask for a number of time slices through the deglaciation of the NH ice sheets.

Response: Why is this strange? We use a different algorithm for the large ice sheets compared to the smaller ones (e.g. the Alps), as described in section 3.1.

With respect to good practice: We already show deglaciation on the north American ice sheet in Figure 6. that is the basis of our validation. It is not clear to us why this section consistently ignored in this review. Additionally the entire data is freely available, so anyone who want to inspect it more thorough is free to do so.

**1335.**

Figure 2. Fine to have a figure for the LGM, but it would be useful to also have a figure some time during the deglaciation. That is when interpolation to a different topography happens and eventual problems with the method could be inspected. I would suggest to document three time slices with figures similar to Fig2 for the LGM, halfway into the deglaciation and present day. Depending on how different they are, one or two could be pushed to a supplement.

**Response: We already show deglaciation on the north American ice sheet in Figure 6. that is the basis of our validation. It is not clear to us why this section consistently ignored in this review? **REVIEWER 2**

The latest version of the manuscript leaves much better impression when compared with the previous one. I appreciate authors' efforts to improve the manuscript based on both reviewers' suggestions. It obviously required a lot of work, but the manuscript now has significantly improved structure and consequently, notably improved readability. Also, a newly added validation chapter crucially contributes to the quality of the manuscript. However, I still find there is a space and need for further improvements, so my observations and suggestions in this regard are as follows:

**Thank you for your comments. We have made the necessary changes you suggested.**

1) The title does not correspond adequately to what is presented in the manuscript. It seems to me that the data set is developed with intention to be used in paleo-ecology (at least the introduction indicates so), therefore, there is no reason not to put that into the title. Also, I would remove "V1.0" from the title, I find it useless. I would rather put "highresolution data set" or even

more specifically: "1-km data set" if that is something what will differentiate it from other potential similar datasets. A suggested title could be:

Response: The omission of v1.0 is certainly possible. I would however not put 'development, validation, and application in paleoecology' into the title. The use of this data is not restricted to paleo-ecology, but might also be useful in other fields.

"CHELSA-TraCE21k high-resolution (or "1-km") data set - downscaled transient temperature and precipitation data since the last glacial maximum – development, validation and application in paleo-ecology", or something similar, maybe shorter. If you want to maintain the current title, then I am afraid you would have to change the bigger part of the introduction.

**Response: See our comment above.**

2) It was really an unfortunate decision to name equally both the data set and the algorithm/model - CHELSA V1.2. Regrettably, I don't see significant progress in clarification in that context throughout the manuscript. Only between the lines 60 and 121, a reader can find the next phrases: "CHELSA V1.2 algorithm", "CHELSA V1.2 climate data set", "CHELSA V1.2 mechanistic downscaling model", "CHELSA V1.2 procedure", "CHELSA downscaling model", "CHELSA V1.2 model". It is just unacceptable and extremely confusing. I find it necessary to add a paragraph, for example, somewhere at the beginning of chapter 2, clarifying that and informing the reader there are the dataset and the algorithm/model with the same name. It has to be very clear. And try to use only 2 words in its description throughout the text, for example

"data set" and "model/algorithm". In addition, once again, please, maintain consistency throughout the text and decide whether you want to use "TraCE21k" or "TraCE-21k".

Response: We tried to be more consistent and called all CHELSA 'data' as data and when we talk about the Algorithm, we consistently now say 'algorithm'.

TraCE-21k is the CCSM3 simulation output at a course resolution. CHELSA-TraCE21k is the downscaled output

This is already consistent in the manuscript

3) In abstract, you say:"High resolution, downscaled climate model data are used in a wide variety of applications across environmental sciences". Then in chapter 6, you start with:"Transient long-term climatic data have a wide range of possible applications". Please, add 2-3 examples where exactly, for example, just continuing the sentence by:"..., such as A, B and C, for example".

Response: We added some studies where the data has been used already.

Transient long-term climatic data have a wide range of possible applications, ranging from population genetics (Leugger et al., 2022; Yannic et al., 2020), community ecology (Staples et al., 2022), to evolutionary biology (Cerezer et al., 2022), just to name a few.

4) Are there other comparable downscaled data sets available on the market? If yes, then, what is the advantage of CHELSA-Trace21k, why is it different/better then the other ones, why it should deserve attention, why is it unique? Please, clarify in a sentence/paragraph, in conclusion, for example.

Response: That is easy to answer: None at 1km.

We added:

**Validations show that CHELSA-TraCE21k V1.0 dataset reasonably represents the distribution the distribution of temperature and precipitation through time at an unpreceded 1km spatial resolution**

5) Before submitting a final version of the manuscript, I suggest a thorough inspection regarding the consistency of the use of terms throughout the manuscript, as well as typing and other errors

Specific comments:

Line 8: High-resolution

Response: Changed

Lines 9-10:

Here we introduce a new, high-resolution dataset, named CHELSA-TraCE21k. It is obtained by downscaling TraCE-21k data, using CHELSA V1.2 algorithm with objective to create global monthly climatologies for temperature and precipitation at 30-arc sec spatial resolution in 100-year time steps for the last 21,000 years.

Response: Changed

Line 18: Validations show that CHELSA-TraCE21k V1.0 dataset reasonably represents the distribution...

Response: Changed

Line 27: GCM states for general circulation model, not global coarser grain=>coarser resolutions

Response: Changed

Line 34: has been bridged, or had to be bridged

**Response: Changed**

Line 59: Here we present paleo-climatic data, downscaled from the CCSM3\_TraCE21k model output (or TraCE21k dataset) to a 30-arc sec. resolution using the CHELSA V1.2 algorithm Response: Changed

Line 67: in various parts of the world from

Response: Changed

Line 68: The TraCE-21k simulation has T31\_gx3v5 resolution (.....)

Response: Changed

Line 70-71: Which resolution??

Please, specify the resolution per model component, for example CAM has resolution 3,75°x3,75°, which is important information for this manuscript, because you are downscaling atmospheric variables

Response: Changed: The TraCE-21k simulation output has a T31\_gx3v5 resolution

Line 76:

It includes mean monthly daily 2m mean, minimum, and maximum temperature ?!

Response: Changed

Line 78: ERA stands for 'ECMWF Re-Analysis'

Response: Changed

Lines 332-335: Figure 2, axis labels not very clear, should be improved

Response: It is clear in the high res. vector graphic so we assume it will be fine in the final publication.

Line 361, 364, 373: RMSE, not RSME

Response: Changed

Lines 444-450: Figure 6, a) and b) missing on the figure; a) is also missing in the legend

**Response: Changed**

Line 455: ...if the transient...

Response: Changed

Line 498: Separate Figure 7 legend from the text below

Response: Changed

---

## Author Response (AR4)

Dear Dr. Rogozhina,

thanks you for your efforts in editing the manuscript. We have now prepared a detailed response to the remaining issues that were raised and edited the manuscript as well as the response to the reviewers as requested. The edited response to the reviewers is attached at the end of this document as we could only upload a single response file.

Please see a detailed response of to your comments below.

On behalf of all co-authors,

Dirk N. Karger

Dear authors,
With this letter, I would like to inform you that I have made a decision to publish your manuscript after minor revisions. The review process related to this manuscript has been extremely rocky and lengthy due to differences in opinions among reviewers, authors, and the editor (I). Even though it is tempting to opt for a quick completion of such a long process, we (reviewer 2 and I) see the need for some final adjustments in both the manuscript and communications with reviewer 1. Our recommendations are outlined below.

In his final report, reviewer 2 admits that the earlier points raised by his review have been mostly addressed, and he only has minor suggestions for the remaining changes in the manuscript included in their recommendations visible to the authors. Following these suggestions, please, do a thorough proof-read of the manuscript in its entirety, including the text, all figure captions, and legends. Please, ensure that the suggested changes are included in the manuscript's title. Furthermore, I have received a more detailed report from reviewer 2 discussing to which extent the criticism in review 1 has been addressed and whether the lack of responses to some major points is justified. I am combining this assessment by reviewer 2 with my own assessment in this final decision. First, we do not feel that the authors have mastered the art of diplomacy required by the official peer-review process. Some of the responses to review 1 are at least indelicate if not arrogant or even borderline rude. I would like to eliminate such communication style from the official review process, even if you disagree with the reviewer's line of argument. Not only does it obstruct the timely review process, since everyone gets irritated and confrontational, but it also undermines the fact that reviewers invest significant time into reading manuscripts and writing their reports. While doing so, they act as allies of the journal (if the review process is fair), with this suggesting that rudeness towards reviewers is also rudeness towards the editor whose choice of reviewers is far from random (you can be absolutely sure in this case). Taken together, these actions can be treated as disrespectful towards the journal as a whole. Although my final decision is generally in favor of the publication, the above issues will have to be corrected as part of the required minor revisions.

We have concluded that the authors have put significant additional work into improvements of their manuscript, following many suggestions of both reviewers. However, we would also like to stress that only due to reviewers' time investment and ideas, this study is now close to publication. At the initial stage, it rather resembled work-in-progress and not a very thorough one, albeit with some potential that justified its initial publication in discussions. I believe that this energy-saving effort and the lack of diplomacy were the main reasons for tensions, which could be avoided altogether if the manuscript were stronger from the start.

Response:      We are sorry that our response has given a wrong impression of the effort the reviewers have devoted to the article. We edited the response to the reviewers for the second major

revision and changed the tone of the responses that might have been irritating. We indicated the changes in a track changed "response to reviewers" for clarity.

Regarding some of the criticism from review 1 that has not been fully addressed, I encourage the authors to look at the first round of reviews to find the comment 2) mentioned by reviewer 1 in the second round (i.e., the interpolation procedure using temperature as a proxy for ice sheet retreat). It is not enough to simply state that you could not find it and thus could not explain why it was not fully addressed.

*We have looked back onto the reviews and change the response:*

*"With respect to the coupling to air-temperatures, we would like to first reiterate our response, that the amount of ice is of course a balance of precipitation fluxes and temperature changes, which can be physically modeled with a numerical model. As this is not feasible at 1km resolution due to computational limitations we use mean annual temperature as a proxy for an interpolation approach.*

*With respect to the validation of the results we would like to refer you to section 4.3 of our manuscript and our more detailed response to your comment on l220. In section 4.3 the results of our glacier interpolations are compared to one of the datasets you suggest (https://doi.org/10.1016/S1571-0866(04)80209-4) using several test statistics. It shows that our interpolations work well, but also have inaccuracies attached to them and specific points in time. "*

*We also added something in the conclusions with respect to this:*

*"The resulting ice cover from this interpolation can, in some areas, only be a few meters thick, not representing real glaciers, but rather spatial autocorrelation artefact of the interpolation approach used (e.g. see Supplementary Figure S1, 13ka BP). Another source of error is that changes in bedrock due to the release of pressure from the melting ice sheets are not yet included in the algorithm. This can potentially result in several hundred meters of bias in affected areas that have not been taken into account in the current version of the algorithm."*

Regarding your response to reviewer 1's comment related to l12, as explained by the reviewer, ICE6G is not derived from an ice sheet model, and you have corrected this misunderstanding but only after the reviewer 1 has mentioned it twice (in the two rounds of reviews!). However, it is still too vague to write "ice sheet data" – please, look it up in the original article for a more precise definition.

*We changed it in the abstract as well as in the main text. In Peltier 2014, ICE5 G is referred to as a "global model of glacial isostasy", with ICE6G_C is the GPS-refined version of it. We hope that this terminology is correctly mentioned in the manuscript now:*

*In the Abstract:*

*Paleo orography at high spatial resolution and for each timestep is created by combining high resolution information on glacial cover from current and Last Glacial Maximum (LGM) glacier databases and interpolations using data from a global model of glacial isostasy (ICE-6G_C)*

*In the main text:*

*2.2 Global model of glacial isostasy: ICE-6G_C (VM5a)*

*We used the output data of the ICE-6G_C (VM5a) (hereafter ICE6G) model as a basis for the extent of the major ice sheets at 1° resolution. ICE6G is a refinement of the ICE-5G (VM2) (hereafter ICE5G) global model of glacial isostasy model (Peltier, 2004) which has been widely used to model the distribution of*

*major ice sheets through time. ICE6G improves ICE5G by applying all available Global Positioning System (GPS) measurements of vertical motion of the crust that constrain the thickness of local ice cover as well as the timing of its removal.  ICE6G explicitly outputs changes in ice thickness of major ice sheets (e.g. the Laurentide ice sheet) from the LGM till today (Argus et al., 2014; Peltier et al., 2015) at 500 year time steps.*

Although I agree with reviewer 1 that GIA-related changes in the topographic forcing, in addition to changes due to sea level fluctuations, would improve the quality of your final dataset (~few hundreds of meters), I understand that this might not be possible at this stage. Even though it should not be that difficult based on my experience, I am ready to pass on this issue.

Response:        We absolutely agree it would indeed improve the dataset to include bedrock changes in the next version of the downscaling model, and it will certainly be the next step in improvement we are planning. We also changed the response to the reviewer here. It now says:

*"Response:        We already replied to his point and pointed out that this might be partly based on a misunderstanding from the reviewers side. We do not change the bedrock topography, and that it is unfortunately not possible with the current algorithm applied. We simply make sure that the orography (terrain elevation above sea level) we use for the downscaling algorithm is actually adjusted to the sea level. If we would not do that, the land surface would not have changed in the last 21 thousand years due to increasing sea levels.*

*The reviewer mentions:  "The idea to correct present-day topography/bathymetry with global sea-level does not make sense to me. Surface elevation over this time scale does not only change due to changing sea-level, also due to isostatic changes of the bedrock."*

*At this point we can unfortunately only mention that we do not have isostatic changes in the bedrock included in the model yet. It would indeed be nice to include them in an updated version, but it would require to develop a transfer function to get from e.g. coarse grid information of changes in bedrock, to 1km high resolution topography. Since we do not have such a function developed yet to integrate into the model we can only mention that this effect is unfortunately not included. We also do not see how this is a major problem but rather a factor that increases the bias of the model and gives room for future improvements. As an example: Most global climate models do not resolve convective precipitation. But that does not mean that these model are fundamentally 'flawed', just because they do not include all possible processes. It rather introduces a bias from the model that, as mentioned correctly by you, should be raised to the reader's attention."*

We also added something in the conclusions with respect to this:

*"The resulting ice cover from this interpolation can, in some areas, only be a few meters thick, not representing real glaciers, but rather spatial autocorrelation artefact of the interpolation approach used (e.g. see Supplementary Figure S1, 13ka BP). Another source of error is that changes in bedrock due to the release of pressure from the melting ice sheets are not yet included in the algorithm. This can potentially result in several hundred meters of bias in affected areas that have not been taken into account in the current version of the algorithm."*

Finally, your statement that reviewer 1 has ignored section 4.3 when talking about the validation against geological evidence is only partly valid. Why on Earth did you pick the ancient dataset of Dyke et al. (2003) to validate your model and why did you do it over North America only? What a strange choice considering the focus of your other analyses on Europe. There are extremely well resolved geochronological datasets for the British Isles, Scandinavia and even the European Alps. I understand

the comments of the authors regarding their Section 4.3 being ignored, but I also emphasize that the way it is handled now is suboptimal from the point of view of the fields of geochronology and geomorphology. You need to at least discuss your choice of a region (and dataset) for validation. Otherwise, it leaves a heavy feeling.

Response:     The choice of Dyke 2004 was mainly to highlight the global extent of the dataset to the reader. Otherwise we felt that, as you mention correctly, the validation would be very euro centric. We are sorry that this led to more confusion than clarity at this point.

Additionally, we added now a supplement that shows the glacial extent from our model compared to Stroeven et. al 2016 (https://doi.org/10.1016/j.quascirev.2015.09.016) and ICE6G. This dataset allows for an additional visual validation of the glacial component in our model. The results from the comparison confirm the results from North America, where over long periods the accuracy of the glacial reconstructions is above 0.8%, but clearly between 9 ka BP and 6 ka BP our model seems to create a larger mismatch towards these two datasets. We added a few new lines about these results (please see the track changed version of the manuscript).

*"Additionally, we used the data from the extent of the ice sheets over Fennoscandia from 22ka BP to 10 ka BP (Stroeven et al., 2016) for all timesteps for which ICE6G data and data from Stroeven et al. (2016) was available. The results (Appendix 1) show similar to the North American ice sheets that the accuracy is relatively high until 10.5ka BP, with a drop in accuracy at 10ka BP. Therefore we assume that the temperature coupling does introduce errors in the time between 10ka BP and 6ka BP as evident from the comparison with the ice sheets of North America and over Fennoscandia."*

We also reformulated our response to the reviewer here:

*"Reviewer 1: There are large data collections available to constrain the LGM-present glacial extent based on geomorphological constraints (e.g. https://doi.org/10.1016/j.quascirev.2015.09.016; https://doi.org/10.1016/S1571-0866(04)80209-4). How does your approximation compare to those reconstructions? Please include a validation for your method.*

*Response:     We included an validation of the glacier reconstructions in section 4.3. The https://doi.org/10.1016/S1571-0866(04)80209-4 you suggest here is Dyke et al. 2004, which we are already using as a basis for our validation in section 4.3. We initially opted for Dyke to also highlight to the reader that the data from the CHELSA-TraCE21k model is global in extents, as otherwise our validation or plausibility test would be very euro-centric. We however added a comparison between our model, Stroeven et. al 2016 (https://doi.org/10.1016/j.quascirev.2015.09.016) and ICE6G and moved it to the Appendix S1"*

Given the above, I would like the authors to walk through the remaining issues mentioned about and find elegant ways to address them at a limited labor cost. Furthermore, I am asking the authors to go back to their responses to review 1 and remove all the irritation-filled comments and overall apply good practices of diplomacy throughout their updated responses to reviewer 1. This effort will pay off handsomely with a manuscript publication.

Kind regards,
Irina

**Response to reviewer 2**

I appreciate your effort to adequately address most of my comments in the latest version of this manuscript. I would only insist on the technically correct title, with hyphenated compound adjectives, addition of the word "downscaled" and use of capital letters for LGM: "CHELSA-TraCE21k — High-resolution (1-km) downscaled transient temperature and precipitation data since the Last Glacial Maximum".

Response: Changed

There are also some tiny details to be corrected such is, for example, in the sentence in the lines 49-52. A would say there is a word "desired" missing in the part : "...25-fold lower than desired computationally efficient simulations of 1 SYPD...", otherwise, it does not make too much sense.

Response: Changed as suggested

I have also noticed some technical errors (repeating words in abstract, chapter 5 missing and similar), therefore I would suggest a thorough inspection of all the text, figures and legends and technical corrections before the final submission.

Response: We went through the manuscript again and checked grammar, spelling, and technical aspects again. There are a few minor changes related to this.

**EDITOR**

Dear authors,

as you have seen, reports of the two referees have deviated in their attitudes and decisions with regards to your article. In response to the criticism in review 1, you have expressed doubts that reviewer 1 has read your manuscript carefully enough and that their criticism is well grounded. I have therefore asked you to prepare a detailed rebuttal letter explaining your grounds for such statements and a revised manuscript highlighting all changes that you have made in response to reviews 1 and 2. Once you have submitted both the letter and the manuscript, I will evaluate your arguments together with an independent reviewer and will make a final decision about the publication or otherwise.
Good luck and let us hope for a positive outcome.

Kind regards,
Irina

Dear Dr. Rogozhina,

Thank you for your work on this manuscript. We have now prepared a point to point response to all concerns and comments by the two reviewers.

**Reviewer 2:** Suggested mainly changes of the text and a more consistent terminology throughout the manuscript. We incorporated all of the comments this reviewer had.

**Reviewer 1:** Is mainly concerned about how we created the orography that we used for the downscaling algorithm. The concern is that we do not include changes in bedrock elevation (which we cannot at this point). Additional the reviewer asks for a validation of the glaciers, which is however, already included as section 4.3 using a dataset the reviewer actually suggests. We include a detailed response to the critique below. Were we thought that the reviewer had a valid point, we incorporated the suggested changes.

We hope that you can follow our reasoning in this respect and deem the manuscript suitable for publication.

Best regards,

On behalf of all co-authors

Dirk Karger

**REVIEWER 1**

General                                                                                                    comments
* * *
I have reviewed an earlier version of this manuscript as REVIEWER 1. While I acknowledge important improvements in the current version of the manuscript, it requires major revisions to make it suitable for publishing. A few of my earlier comments have not been addressed, so I am reiterating them once more.

With the given documentation, it is still not possible to judge if the applied methodology is flawed. This

concerns 1) the way only sea level (and not bedrock changes) are used to calculate elevation changes and 2) the interpolation procedure using temperature as a proxy for ice sheet retreat. I have suggested possibilities to validate the methods with available reconstructions, which I consider fundamental to support the choices presented in this paper.

Response:     Thank you for your comments and efforts in reviewing our manuscript a second time. It is unfortunate that our previous explanation has not been clear enough regarding the interpolation procedure of the glaciers despite the new and additional figures to explain it step by step. Our model does not include bedrock changes but only sea level changes incorporated. We might include this effect in a newer version of the model to improve it further. We are aware that this model has limitations and cannot be perfect in every regard.

With respect to the coupling to air-temperatures, we would like to first reiterate our response, that the amount of ice is of course a balance of precipitation fluxes and temperature changes, which can be physically modeled with a numerical model. As this is not feasible at 1km resolution due to computational limitations we use mean annual temperature as a proxy for an interpolation approach.

With respect to the validation of the results we would like to refer you to section 4.3 of our manuscript and our more detailed response to your comment on l220. In section 4.3 the results of our glacier interpolations are compared to one of the datasets you suggest (https://doi.org/10.1016/S1571-0866(04)80209-4) using several test statistics. It shows that our interpolations work well, but also have inaccuracies attached to them and specific points in time.

Please find a detailed response to the specific concerns raised below.

Specific comments
* * *
l12.
Repeating my earlier comment: ICE6G is not a dynamic ice sheet model. I have looked up the description for you: "the ICE-6G_C (VM5a) model that is under discussion in this paper is based upon the 'GIA only' methodology. In this methodology the ice thickness history as a function of position is simply adjusted iteratively in order to satisfy all of the available constraints". Please reformulate.

Response:     Changed to: "…and interpolations using ice sheet data (ICE6G)…"

l26.
Here you state "climatic conditions at spatial resolutions < 1 km", but later (e.g. 48) it is "~1km". Should be made consistent.

Response:     Changed

l27.
"run at much coarser grains" --> "run at much coarser resolution"

Response:     Changed

l34.
"has be bridged" --> "has been bridged"

Response:        Changed

l44.
"on earth" --> "on Earth"

Response:        Changed

l48.
It is somewhat trivial to state that 0.043 SYPD is 25-fold lower than 1 SYPD. What would be an example of a typical 1 SYPD simulation?

Response:        It is unclear what is asked here for. SYPD is a measurement of computational efficiency.

l60.
Repeating comments for "time steps of 100 years from 21k-BP to 1990"
Here and elsewhere, this should be written as "21 kyr BP"
BP typically refers to years before 1950. If you go in steps of 100 from 21 kyr BP forward, you never end up at 1990. Either you used a different time step than 100 years, or a different starting point. That is why you have to modify this statement.

Response:        Changed to ka BP

l60.
(TraCE-21k) should be defined the first time you use it (outside of the abstract).

Response:        Changed

l64.
It would be good to add some information about what topography and ice sheet boundary condition (topography, mask, albedo) was used to produce this simulation. Is the land-sea mask constant? Is the topography changing? If yes, is the topography change consistent with the ICE6G_C reconstruction?

Response:        This is in detail given in:

He, F.: Simulating Transient Climate Evolution of the Last Deglaciation with CCSM3, PhD - Thesis, University of Wisconsin Madison, Madison, WC, USA, 171 pp., 2011.

Liu, Z., Otto-Bliesner, B. L., He, F., Brady, E. C., Tomas, R., Clark, P. U., Carlson, A. E., Lynch-Stieglitz, J., Curry, W., Brook, E., Erickson, D., Jacob, R., Kutzbach, J., and Cheng, J.: Transient Simulation of Last Deglaciation with a New Mechanism for Bølling-Allerød Warming, Science, 325, 310–314, https://doi.org/10.1126/science.1171041, 2009.

We did not run the CCSM3 TraCE-21k simulations by ourselves, so we think it would not be appropriate to repeat the methods from He et al. and Liu et al. here, which would be outside of the scope of the paper.

l69.
Is "DGVM" the name of the specific model or the shorthand for all dynamic global vegetation models (as line 71-72 may suggest)? Clarify!

Response:    as stated:    "…dynamic global vegetation model (DGVM)…"

l70.
Is the "land [...] model" the same as the vegetation model? If not, introduce the land model here.

Response:    It's the land component of CCSM3. We tried to clarify:

"…The TraCE-21k simulation was calculated at a T31_gx3v5 resolution (Otto-Bliesner et al., 2006) using a coarse resolution dynamic global vegetation model (DGVM). The coupled atmosphere-ocean model in CCSM3 is based on  the Community Atmospheric Model 3 (CAM3), on 26 vertical hybrid coordinate levels. The land and atmosphere components in CCSM3 in the TraCEF-21k simulations uses the same resolution. The parameterizations of the DGVM are largely based on the Lund-Potsdam-Jena (LPJ)-DGVM. The ocean model in CCSM3 uses the NCAR (National Center for Atmospheric Research) version of the Parallel Ocean Program (POP) with 25 vertical levels and the sea ice model is the NCAR Community Sea Ice Model (CSIM)…"

l76.
What is "mean monthly daily 2m mean"? Maybe "monthly mean, minimum and maximum temperature and precipitation fields"?

Response:    Changed

l78.
You call it "mechanistic climate downscaling". I am familiar with terms statistical and dynamical downscaling. Which category does yours belong to?

Response:    None of the above. It is a hybrid model, that has mechanistic components, but also statistical ones. Another term we commonly use is topographic downscaling.

Changed to:    topographic downscaling

l84.
Remove "and dynamics". ICE6G is not a dynamic ice sheet model.

Response:    Changed

l85.
"from the Last Glacial Maximum"
What is the first available year in the time series?

Response:    26kyr BP? We however only start our downscaling at the LGM as stated.

l87.
LGM is already defined. Remove "(LGM)"

Response:    We removed 'Last glacial maximum' and keep the abbreviation

l88.
LGM is already defined. Modify description.

Response:    We removed 'Last glacial maximum' and keep the abbreviation

l89.

How "up-to-date" is this dataset from more than 10 years ago?
Suggest to reformulate.

Response:        deleted 'up to date'

l95.
What time does "the 'current' extent of the glaciers" refer to? I understand that this may not be clearly defined by the dataset providers (late 90s, early 2000 maybe), but you are assigning it eventually to a certain time in your modelling (maybe 1990?). This should be mentioned here or elsewhere as in "We are assigning the dataset to the year xxx in our modelling". The same applies to all the other data sources (GMTED, GEBCO), which may be best done by a summary statement in the end.

Response:        Unfortunately the GLIMS database does not give a reference year. We therefore use as referring to the reference period (1950-1990).

L105.
"we keep as land altimetric data that of the CHELSA V1.2 procedure"
I didn't find a description what topography is used in CHELSA. Should be added in 2.2.

Response:        Page 3, Line 13 in Karger et al. 2017, Sci. Dat. "…the Global Multi-resolution Terrain Elevation Data 2010 (GMTED2010)…". or Figure 1.

We included it now:

Although GEBCO also includes land surface altitude, we only use it for the oceans, and we keep as land altimetric data that of the CHELSA V1.2 algorithm (that being GMTED2010) to maintain comparable topography at the land surface.

Section 2 in general
Since this has become a long list of rather short subsections, it could be an idea to display the information about the different datasets in a table (name, description, time coverage, reference, ...) complemented with a summary paragraph.

Response:        This is a matter of personal preferences and we actually prefer to keep it as it is.

l117.
"As the orography at different time steps between 21k BP and current times is not available"
I think that ICE6G_C would give you that information. Maybe add "... at the high resolution required for our downscaling method" to make sense of this statement.

Response:        Changed to:      "…As the orography at different time steps between 21ka BP and current times is not available at the high resolution required for the CHELSA algorithm…"

Figure 1, l124.
Terminology. I am not an expert on this, but it seems that the distinction made here between topography (as relative to present day sea-level) and orography (relative to current global sea-level) is not supported by common definitions of the two terms. Also, I could not find a description of

bathymetry in the ocean and surface elevation over land in one generalised term. I would suggest to using your own symbols and describing what they mean, rather than using established terms that mean something else.

Response:        There seems to be a misunderstanding: Orography generally refers to terrain above water (including glacier surfaces), topography can also include terrain under water, bathymetry contains terrain under water.  This is the same terminology as used in ICE6G_C (e.g. orog and topo) for example.

I doubt that the combination of GMTED2010 and GEBCO really gives you bedrock topography (upper right in figure 1). I think GMTED2010 provides surface elevation over glaciated areas (Greenland/Antarctica), which is the upper ice surface. The bedrock topography is a few thousand meters below that. The same applies to 'bedrock' in the downstream box 'bedrock orography' to the left.

Response:        That is correct, it gives the bedrock for non-glaciated areas. For the approach used here, this is however does not constitute a problem, since we are interpolating between past and current glacier extent. We clarified this by adding:

*"…To create a bedrock orography $e_t^{bed}$ (i.e. topography adjusted for sea level without glaciers except for currently glaciated areas)…"*

The idea to correct present-day topography/bathymetry with global sea-level does not make sense to me. Surface elevation over this time scale does not only change due to changing sea-level, also due to isostatic changes of the bedrock. In the periphery of the ice sheets where it may be most important for your biological application, the bedrock change may well be the dominant signal. I see in l169 that you acknowledge that bedrock changes are not taken into account. But why not?. ICE6G_C will give you a consistent set of data for sea level and bedrock elevation, in addition to ice thickness. In fact, the bedrock change is likely the most reliable output of ICE6G_C, because ice thickness is prescribed to get the right loading history. Why are you not using it? If you think your sea-level correction method gives a better representation of surface elevation, you should at least compare your results (with appropriate figures) against ICE6G_C. This will show if your method is an appropriate approximation for the full solution. My intuition is that surface elevation will be off by a few hundred meters in proximity of an ice sheet. If I am overlooking something obvious here, please explain better in the manuscript why you chose to not use the full information provided by ICE6G_C.

Response:        We already replied to his point and pointed out that this might be partly based on a misunderstanding from the reviewers side. We do not change the bedrock topography, and that it is unfortunately not possible with the current algorithm applied. We simply make sure that the orography (terrain elevation above sea level) we use for the downscaling algorithm is actually adjusted to the sea level. If we would not do that, the land surface would not have changed in the last 21 thousand years due to increasing sea levels.

The reviewer mentions:  "The idea to correct present-day topography/bathymetry with global sea-level does not make sense to me. Surface elevation over this time scale does not only change due to changing sea-level, also due to isostatic changes of the bedrock."

At this point we can unfortunately only mention that we do not have isostatic changes in the bedrock included in the model yet. It would indeed be nice to include them in an updated version, but it would require to develop a transfer function to get from e.g. coarse grid information of changes in bedrock, to 1km high resolution topography. Since we do not have such a function developed yet to

integrate into the model we can only mention that this effect is unfortunately not included. We also do not see how this is a major problem but rather a factor that increases the bias of the model and gives room for future improvements. As an example: Most global climate models do not resolve convective precipitation. But that does not mean that these model are fundamentally 'flawed', just because they do not include all possible processes. It rather introduces a bias from the model that, as mentioned correctly by you, should be raised to the reader's attention.

In the conclusions we added:
*"…The resulting ice cover from this interpolation can, in some areas, only be a few meters thick, not representing real glaciers, but rather spatial autocorrelation artefact of the interpolation approach used (e.g. see Supplementary Figure S1, 13ka BP). Another source of error is that changes in bedrock due to the release of pressure from the melting ice sheets are not yet included in the algorithm. This can potentially result in several hundred meters of bias in affected areas that have not been taken into account in the current version of the algorithm…"*

l128.
The following description still misses clear motivations for why things are done the way they are done. You should start by making clear what information you have (e.g. present day seabed and land topography), what you are trying derive, and how you are making approximations/interpolations to get there.

Response:      From the comment as given it is not entirely sure to us what is missing here. We tried to do exactly what has been suggested here. First we state the information we have:

"The first step in estimating the paleo-orography was carried out for the LGM (21k BP). For this time point, both estimates of glacial extents from Ehlers et al., 2011 and estimates of glacier thickness from ICE6G exist. "

, after that we state what we want to derive:

"We first combined the topographic information from GMTED2010 on land, and that of GEBCO into a bedrock topography that provides the current bedrock topography e_c^topo (including current day glaciers, see ff.)"

In figure 1 and figure 2 we show the process in a graphically. Additionally the equations are all given to allow to better assess the way it is done.

l130.
I think from "We first combined ..." you are no longer at the LGM. That would be good to make clear. E.g. with "that provides the surface elevation e at the present day"

Response:      Changed:        "…We first combined the topographic information from GMTED2010 on land, and that of GEBCO into a bedrock topography that provides the current bedrock topography $e_c^{topo}$ (including current day glaciers, see ff.)…"

l136.
Back to LGM? Confusing!
I think from l136-l147 (maybe even l149) you are describing how to produce a smooth ice surface elevation for the LGM on a high-resolution grid. Could be good to say that upfront.
The then following transition into a time-dependent estimate is confusing to me, because we don't know at this point how the interpolation between LGM and present will work. Maybe it would be better to leave the time dependence out of this for now?

**Response:** We describing first how we create the initial high resolution ice surface at the LGM (t=0) and then how we use this as a basis for the next timestep. (t0 … t*n*). Since this is a iterative process over time removing the time dependence here would not be correct.

l170.
Without any validation/comparison against reconstructed glacial extents from the literature, it is not possible to judge whether this interpolation approach is an innovative method or a bad idea. See also comment l220.

**Response:** At this point we would like to refer to section 4.3 "Validation of glacier extent between 18ka BP and 1ka BP" and our response to the comment in l220.

l171.
"As high-resolution estimates of glacial surface elevation are not available for timesteps t other than the LGM"
For the LGM you had Ehlers2011 to delineate glacier extents, but the resolution of the surface elevation data is the same for all time steps in ICE6G_C, isn't it?.

**Response:** Yes, it is all the same: 1°.

l173.
"at each time step $t \neq 0$"
But also not at present day, right?

**Response:** Correct. Changed to:

"…at each time step $t \neq 0$ and t = 221…"

l181.
"resampled to a 0.5° grid resolution"
Can you motivate the need for this intermediate grid? Why not interpolate one of the products to the grid of the other?

**Response:** This is the same resolution as in the CHELSA V1.2 algorithm. We kept it constant with the algorithm.

**We added:** "…The resolution of 0.5° follows the same procedure as used in CHELSA 1.2 (Karger et al. 2017)…"

l220.
There are large data collections available to constrain the LGM-present glacial extent based on geomorphological constraints (e.g. https://doi.org/10.1016/j.quascirev.2015.09.016; https://doi.org/10.1016/S1571-0866(04)80209-4). How does your approximation compare to those reconstructions? Please include a validation for your method.

**Response:** We included an validation of the glacier reconstructions in section 4.3. We also show the deglaciation on the north American ice sheet in Figure 6. that forms the basis of our validation. The https://doi.org/10.1016/S1571-0866(04)80209-4 you suggest here is Dyke et al. 2004, which we are already using as a basis for our validation in section 4.3. We opted for Dyke to also highlight to the reader that the data from the CHELSA-TraCE21k model is global in extents, as otherwise our validation or plausibility test would be very euro-centric.

l240.
'idiosyncratic'
I have commented this before and supposedly you had changed it. Here it is again.

Changed to: "…The CHELSA V1.2 algorithm assumes that orography is one of the main drivers of precipitation…"

l299.
Could you motivate the parameter choices for c and h? Have you tried other values? Are they taken from past experience with the model or from published values?

Response: They come from the tuning of CHELSA V1.2. See Karger et al. 2017, Sci. Dat.

l304.
Remind us what reference period the bias correction is calculated over and that you are using an intermediate grid.

Response: 1980-1990. Added

l331.
"tas being interchangeable for tasmax and tasmin in Eq. 22 and Eq. 23"
This doesn't work, because you have already defined tas = (tasmax+tasmin)/2 in line 323. Need to find another symbol.

Response: Clarified to:

"…The downscaling of monthly near surface air temperatures (*tas, tasmax, tasmin*) follows the methods described in 3.2.2., with the only difference that instead of mean annual temperature, *tasmax* and *tasmin* are used, where *tas* = (*tasmax*+*tasmin*)/2. The temperatures have again first been bias corrected using:

$$\Delta tasmax_m = tasmax^{obs}_{cur_m} - tasmax^{mod}_{cur_m} \tag{22}$$

$$\Delta tasmin_m = tasmin^{obs}_{cur_m} - tasmin^{mod}_{cur_m} \tag{23}$$

and:

$$tasmax^{cor}_{m_t} = tasmax^{obs}_{cur_m} - \Delta tasmax_m \tag{24}$$

$$tasmin^{cor}_{m_t} = tasmin^{obs}_{cur_m} - \Delta tasmin_m \tag{25}$$

with *m* being the respective month of the year, in Eq. 22 - Eq. 25…."

l335.
Figure2. It seems strange to switch region in the middle of the description from e) to f). It would be useful to continue with the first region to the end and add another figure for the second region if needed. For the validation of interpolated ice extent, it would also be good practice to show your reconstructed ice mask for a number of time slices through the deglaciation of the NH ice sheets.

Response:     We use a different algorithm for the large ice sheets compared to the smaller ones (e.g. the Alps), as described in section 3.1. which is why we include a switch in region here.

l335.
Figure2. Fine to have a figure for the LGM, but it would be useful to also have a figure some time during the deglaciation. That is when interpolation to a different topography happens and eventual problems with the method could be inspected. I would suggest to document three time slices with figures similar to Fig2 for the LGM, halfway into the deglaciation and present day. Depending on how different they are, one or two could be pushed to a supplement.

Response:     We already included an validation of the glacier reconstructions in section 4.3 with respective time slices for Dyke et al. 2004. We initially opted for Dyke to also highlight to the reader that the data from the CHELSA-TraCE21k model is global in extents, as otherwise our validation or plausibility test would be very euro-centric. We however added a comparison between our model, Stroeven et. al 2015 (https://doi.org/10.1016/j.quascirev.2015.09.016) and ICE6G and moved it to the Appendix S1 which shows the timesteps for which ICE6G, CHELSA-TraCE21k, and data from Stroeven et. al 2015 is available.

The results from the comparison confirm the results from North America, where over long periods the accuracy of the glacial reconstructions is above 0.8%, but clearly between 9 ka BP and 6 ka BP our model seems to create a larger mismatch towards these two datasets. We added a few new lines about these results.

**REVIEWER 2**
The latest version of the manuscript leaves much better impression when compared with the previous one. I appreciate authors' efforts to improve the manuscript based on both reviewers' suggestions. It obviously required a lot of work, but the manuscript now has significantly improved structure and consequently, notably improved readability. Also, a newly added validation chapter crucially contributes to the quality of the manuscript. However, I still find there is a space and need for further improvements, so my observations and suggestions in this regard are as follows:

Thank you for your comments. We have made the necessary changes you suggested.

1)     The title does not correspond adequately to what is presented in the manuscript. It seems to me that the data set is developed with intention to be used in paleo-ecology (at least the introduction indicates so), therefore, there is no reason not to put that into the title. Also, I would remove "V1.0" from the title, I find it useless. I would rather put "highresolution data set" or even more specifically: "1-km data set" if that is something what will differentiate it from other potential similar datasets. A suggested title could be:

Response:     The omission of v1.0 is certainly possible. I would however not put 'development, validation, and application in paleoecology' into the title. The use of this data is not restricted to paleo-ecology, but might also be useful in other fields.

"CHELSA-TraCE21k high-resolution (or "1-km") data set - downscaled transient temperature and precipitation data since the last glacial maximum – development, validation and application in paleo-ecology", or something similar, maybe shorter. If you want to maintain the current title, then I am afraid you would have to change the bigger part of the introduction.

Response:     See our comment above.

2)     It was really an unfortunate decision to name equally both the data set and the algorithm/model - CHELSA V1.2. Regrettably, I don't see significant progress in clarification in that

context throughout the manuscript. Only between the lines 60 and 121, a reader can find the next phrases: "CHELSA V1.2 algorithm", "CHELSA V1.2 climate data set", "CHELSA V1.2 mechanistic downscaling model", "CHELSA V1.2 procedure", "CHELSA downscaling model", "CHELSA V1.2 model". It is just unacceptable and extremely confusing. I find it necessary to add a paragraph, for example, somewhere at the beginning of chapter 2, clarifying that and informing the reader there are the dataset and the algorithm/model with the same name. It has to be very clear. And try to use only 2 words in its description throughout the text, for example
"data set" and "model/algorithm". In addition, once again, please, maintain consistency throughout the text and decide whether you want to use "TraCE21k" or "TraCE-21k".

Response:        We tried to be more consistent and called all CHELSA 'data' as data and when we talk about the Algorithm, we consistently now say 'algorithm'.

TraCE-21k is the CCSM3 simulation output at a course resolution.
CHELSA-TraCE21k is the downscaled output

This is already consistent in the manuscript

3)        In abstract, you say:"High resolution, downscaled climate model data are used in a wide variety of applications across environmental sciences". Then in chapter 6, you start with:"Transient long-term climatic data have a wide range of possible applications". Please, add 2-3 examples where exactly, for example, just continuing the sentence by:"..., such as A, B and C, for example".

Response:        We added some studies where the data has been used already.

Transient long-term climatic data have a wide range of possible applications, ranging from population genetics (Leugger et al., 2022; Yannic et al., 2020), community ecology (Staples et al., 2022), to evolutionary biology (Cerezer et al., 2022), just to name a few.

4)        Are there other comparable downscaled data sets available on the market? If yes, then, what is the advantage of CHELSA-Trace21k, why is it different/better then the other ones, why it should deserve attention, why is it unique? Please, clarify in a sentence/paragraph, in conclusion, for example.

Response:        That is easy to answer: None at 1km.

We added:

Validations show that CHELSA-TraCE21k V1.0 dataset reasonably represents the distribution the distribution of temperature and precipitation through time at an unpreceded 1km spatial resolution

5)        Before submitting a final version of the manuscript, I suggest a thorough inspection regarding the consistency of the use of terms throughout the manuscript, as well as typing and other errors

Specific comments:

Line 8: High-resolution

Response:        Changed

Lines 9-10:
Here we introduce a new, high-resolution dataset, named CHELSA-TraCE21k. It is obtained by downscaling TraCE-21k data, using CHELSA V1.2 algorithm with objective to create global monthly

climatologies for temperature and precipitation at 30-arc sec spatial resolution in 100-year time steps for the last 21,000 years.

Response:      Changed

Line 18:
Validations show that CHELSA-TraCE21k V1.0 dataset reasonably represents the distribution...

Response:      Changed

Line 27:
GCM states for general circulation model, not global coarser grain=>coarser resolutions

Response:      Changed

Line 34:
has been bridged, or had to be bridged

Response:      Changed

Line 59:
Here we present paleo-climatic data, downscaled from the CCSM3_TraCE21k model output (or TraCE21k dataset) to a 30-arc sec. resolution using the CHELSA V1.2 algorithm
Response:      Changed

Line 67:
in various parts of the world from

Response:      Changed

Line 68:
The TraCE-21k simulation has T31_gx3v5 resolution (…..)

Response:      Changed

Line 70-71:
Which resolution??
Please, specify the resolution per model component, for example CAM has resolution 3,75°x3,75°, which is important information for this manuscript, because you are downscaling atmospheric variables

Response:      Changed: The TraCE-21k simulation output has a T31_gx3v5 resolution

Line 76:
It includes mean monthly daily 2m mean, minimum, and maximum temperature ?!

Response:      Changed

Line 78:
ERA stands for 'ECMWF Re-Analysis'

Response:      Changed

Lines 332-335:
Figure 2, axis labels not very clear, should be improved

Response:        It is clear in the high res. vector graphic so we assume it will be fine in the final publication.

Line 361, 364, 373:
RMSE, not RSME

Response:        Changed

Lines 444-450:
Figure 6, a) and b) missing on the figure; a) is also missing in the legend

Response:        Changed

Line 455:
…if the transient…

Response:        Changed

Line 498:
Separate Figure 7 legend from the text below

Response:        Changed